# Agent-based modelling of reactive vaccination of workplaces and schools against COVID-19

Benjamin Faucher[1], Rania Assab[1], Jonathan Roux[2], Daniel Levy-Bruhl[3], Cécile Tran Kiem [4,5], Simon Cauchemez [4], Laura Zanetti[6], Vittoria Colizza [1,7], Pierre-Yves Boëlle[1] & Chiara Poletto [1 ✉]

With vaccination against COVID-19 stalled in some countries, increasing vaccine accessibility and distribution could help keep transmission under control. Here, we study the impact of reactive vaccination targeting schools and workplaces where cases are detected, with an agent-based model accounting for COVID-19 natural history, vaccine characteristics, demographics, behavioural changes and social distancing. In most scenarios, reactive vaccination leads to a higher reduction in cases compared with non-reactive strategies using the same number of doses. The reactive strategy could however be less effective than a moderate/high pace mass vaccination program if initial vaccination coverage is high or disease incidence is low, because few people would be vaccinated around each case. In case of flare-ups, reactive vaccination could better mitigate spread if it is implemented quickly, is supported by enhanced test-trace-isolate and triggers an increased vaccine uptake. These results provide key information to plan an adaptive vaccination rollout.

[1] Sorbonne Université, INSERM, Institut Pierre Louis d'Epidémiologie et de Santé Publique, Paris, France. [2] Univ Rennes, EHESP, CNRS, ARENES—UMR 6051, F-35000 Rennes, France. [3] Santé Publique France, Saint Maurice, France. [4] Mathematical Modelling of Infectious Diseases Unit, Institut Pasteur, Université de Paris, UMR2000, CNRS, Paris, France. [5] Collège Doctoral, Sorbonne Université, Paris, France. [6] Haute Autorité de Santé, Saint-Denis, France. [7] Tokyo Tech World Research Hub Initiative (WRHI), Tokyo Institute of Technology, Tokyo, Japan. ✉email: chiara.poletto@inserm.fr

Vaccination against SARS-CoV-2 has changed the course of the COVID-19 pandemic due to the high efficacy of available vaccines in preventing infection and severe disease. Yet, several months into the vaccination campaign, vaccine uptake remained below official targets in many Western countries due to logistical issues, vaccine accessibility and/or hesitancy. As of Fall 2021, less than 60% of the population in the United States and Europe was fully vaccinated[1]. With intense virus circulation still ongoing in many regions of the world due to the Delta variant and the threat posed by emerging variants, it is important to investigate whether vaccine use could improve with adaptive delivery. Indeed, offering vaccination to individuals who were exposed to the virus allows targeting those at higher risk of infection and, furthermore, might help overcome barriers to vaccination[2,3] since vaccine-hesitant people are more likely to accept vaccination when the perceived risk of infection is higher[4].

Redirecting vaccine supplies to geographic areas of highest incidence (or hotspot vaccination) is already part of the plans in some European countries and was implemented to combat the emergence of the Delta variant[2]. Other reactive vaccination schemes are possible, such as ring vaccination that targets contacts of confirmed cases or contacts of those contacts, or vaccination in workplaces or schools where cases have been detected. This could potentially improve the impact of vaccination by preventing transmission where it is active and even enable the efficient management of flare-ups. For smallpox or Ebola fever, ring vaccination has proved effective in rapidly curtailing outbreaks[5–8]. However, the experience of these past epidemics cannot be transposed directly to COVID-19 due to the many differences in the infection characteristics and epidemiological context. For example, COVID-19 cases are infectious a few days before symptom onset[9] but often detected a few days after. This gives time to infect their direct contacts and thwarts ring vaccination. Vaccinating an extended network of contacts, as could be done with the vaccination of whole workplaces or schools, would have a larger impact, especially if adopted in combination with strengthened protective measures to slow down transmissions, such as masks, physical distancing and contact tracing. This could be feasible in many countries, leveraging the established test-trace-isolate (TTI) system that enables prompt detection of clusters of cases to decide where vaccines should be deployed. Properly assessing the interest of reactive vaccination therefore requires a detailed examination of the interactions of vaccine characteristics, the pace of vaccination, COVID-19 natural history, case detection practises and overall changes in population behaviour.

We therefore extend an agent-based model that has been previously described[10] to quantify the impact of a reactive vaccination strategy targeting workplaces, universities and 12+ years old in schools where cases have been detected. We compare the impact of reactive vaccination with non-reactive vaccination targeting similar settings or with mass vaccination, and test these strategies alone and in combination. We explore differences in vaccine availability and logistical constraints, and assess the influence of the dynamic of the epidemic and different stages of the vaccination campaign.

## Results

**Mass vaccination, targeted and reactive vaccination strategies.** We extended a previously described SARS-CoV-2 transmission model[10] to simulate vaccine administration alongside other interventions—i.e. contact tracing, teleworking and social restrictions. Following similar approaches[11–14], the model is stochastic and individual-based. It takes as input a synthetic population reproducing demographic and social-contact data, workplace sizes and school types (Fig. 1a) of a typical medium-sized French town (117,492 inhabitants). Contacts are described as a dynamic multilayer network[10] (Fig. 1b).

We assumed that the vaccine reduced susceptibility, quantified by the vaccine effectiveness $VE_S$, and symptomatic illness after infection, quantified by $VE_{SP}$[15] (Fig. 1c). We considered a vaccination strategy based on the Cominarty vaccine[16] which is very suitable for reactive vaccination given efficacy, only 3 weeks between the two doses and wide availability. We described the vaccine-induced protection with respect to the Delta variant—i.e. the dominant variant as of Fall 2021. Real-life estimates are heterogeneous, reflecting the complex interplay between the timing of Delta introduction in the population, the co-circulation of other variants, waning of immunity and differential impact by age. In the baseline scenario we considered vaccine effectiveness levels in the middle of the range of estimates provided in a systematic review[17]. We used a three-week interval between doses as in the vaccine trial[16]. For vaccine protection, we conservatively assumed that there was no protection in the 2 weeks after the first inoculation, followed by intermediate protection until 2 weeks after the second dose ($VE_{S,1} = 48\%$ and $VE_{SP,1} = 53\%$, see additional details in the Supplementary Table 2) and maximum protection afterwards ($VE_{S,2} = 70\%$ and $VE_{SP,2} = 73\%$), 5 weeks after the first dose. The maximum protection values are close to the estimates obtained in a meta-analysis for Delta, all vaccines combined[18]. Lower and higher vaccine effectiveness are also explored.

In the baseline scenario we parametrised the epidemiological context assuming that 32%[19,20] of the population was fully immune to the virus due to the previous infection. Initial incidence was moderate/high, i.e. ~160 clinical cases weekly per 100,000 inhabitants, and the reproductive ratio was $R = 1.6$, in the range of values estimated for the Delta wave of summer 2021[1,21]. We modelled the baseline TTI policy after the French situation, allowing 3.6 days on average from symptoms onset to detection and 2.8 average contacts detected and isolated per index case[22] (Fig. 1d). We assumed that 50% of clinical cases and 10% of subclinical cases were detected, leading to an overall detection rate of ~25%[20,23]. Social restrictions were modelled assuming 10% of individuals were doing teleworking and contacts in the community were reduced by 5% (see Methods).

We then modelled vaccination targeting all adults older than 12 years old with baseline vaccine uptake—set to 80% in the 12-65 years old and 90% in the over 65 years old[24]. We assumed that priority risk groups (e.g. elderlies) had already been vaccinated up to that level at the start[25–28]. We modelled three non-reactive vaccination strategies in the general population, where vaccination was carried out up to the maximum number of doses available each day at random (i) in the whole population (mass) or (ii) in schools sites (school locations, described below) or (iii) in workplaces/universities (workplaces/universities). In the school locations vaccination, we assumed vaccine sites were set up in schools to vaccinate pupils and their parents/siblings over the age of 12[29]. Then, we modelled a reactive vaccination strategy, where the detection of a case thanks to TTI triggered the vaccination of household members and those in the same workplace or school (Fig. 1d). In this scenario, a baseline delay of 2 days on average was assumed between the detection of the case and starting vaccination to account for logistical issues—i.e. ~5.6 days on average from the index case's symptoms onset. In the baseline scenario, we assumed vaccine uptake in the context of reactive vaccination to be the same as in non-reactive vaccination. The impact of each strategy was assessed by comparison with a reference scenario, where no vaccination campaign is conducted during the course of the simulation and vaccination coverage remains at its initial level.

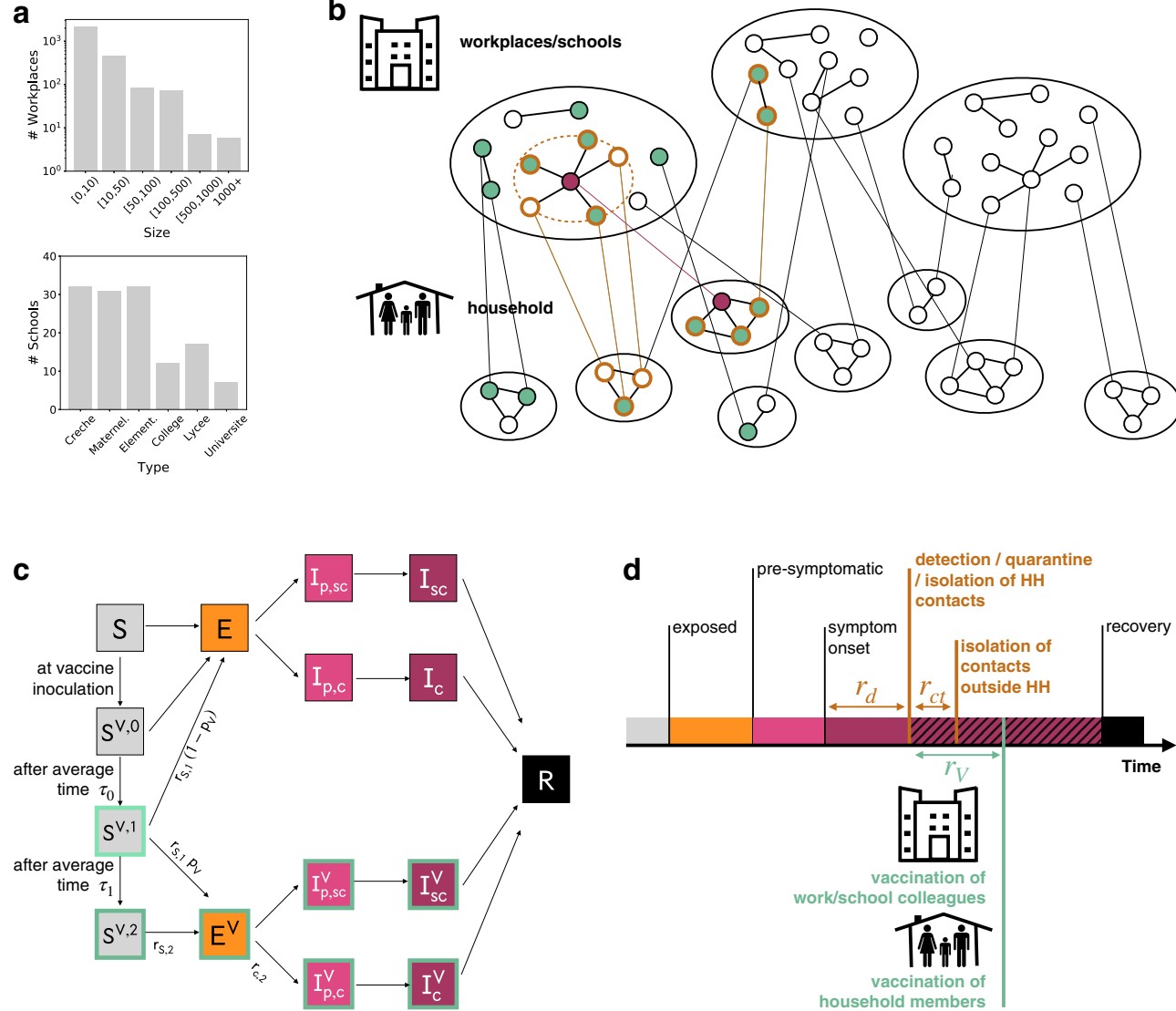

**Fig. 1 Modelling reactive vaccination. a** Distribution of workplace size and of school type for the municipality of Metz (Grand Est region, France), used in the simulation study. **b** Schematic representation of the population structure, the reactive vaccination and contact tracing. The synthetic population is represented as a dynamic multi-layer network, where layers encode contacts in household, workplace, school, community and transport. In the figure, school and workplace layers are collapsed and community and transport are not displayed for the sake of visualisation. Nodes repeatedly appear on both the household and the workplace/school layer. The identification of an infectious individual (in purple in the figure) triggers the detection and isolation of his/her contacts (nodes with orange border) and the vaccination of individuals attending the same workplace/school and belonging to the same household who accept to be vaccinated (green). **c** Compartmental model of COVID-19 transmission and vaccination. Description of the compartments is reported on the Methods section. **d** Timeline of events following infection for a case that is detected in a scenario with reactive vaccination. For panels **c**, **d** transition rate parameters and their values are described in the Methods and in the Supplementary Information.

In Fig. 2a–h we used the baseline parameters values presented above but varied the initial vaccination coverage level for comparison. We first considered the case of low vaccination coverage, i.e. ~30% over the population—with 15% of the [12,60] group and 90% of the 60+ group—as seen in some countries in Eastern Europe and some US counties in the fall of 2021[1,30]. For non-reactive strategies, the vaccination pace ranged between 100 and 500 first doses per 100,000 inhabitants per day. The vaccination pace in Western countries roughly fell within these extremes for the majority of the vaccination campaign, with lower values in general reached around the beginning and the end, due to delivery issues at the beginning, and difficulty in overcoming barriers to vaccination at the end[1]. For the reactive strategy, vaccine deployment is triggered by detected cases, therefore the number of doses used and the number of places where these doses

are administered depends on the epidemic situation. Figure 2a shows the relative reduction in the attack rate after 2 months as a function of the number of first daily doses and Fig. 2b compares the incidence profiles under different strategies with the same number of vaccine doses. The mass, school location and workplaces/universities strategies have a similar impact on the epidemic. They lead to a reduction between 2.7% and 3% of the attack rate with 100 first doses per 100,000 inhabitants administered each day, and between 13% and 15% with 500 per 100,000 inhabitants. Among the three strategies, the reduction produced by mass vaccination was slightly lower. This is because the strategies are compared at the same number of daily vaccine doses and, in workplaces/universities and school locations, these doses were directed to a more active population—working population, or population living in large households—with a

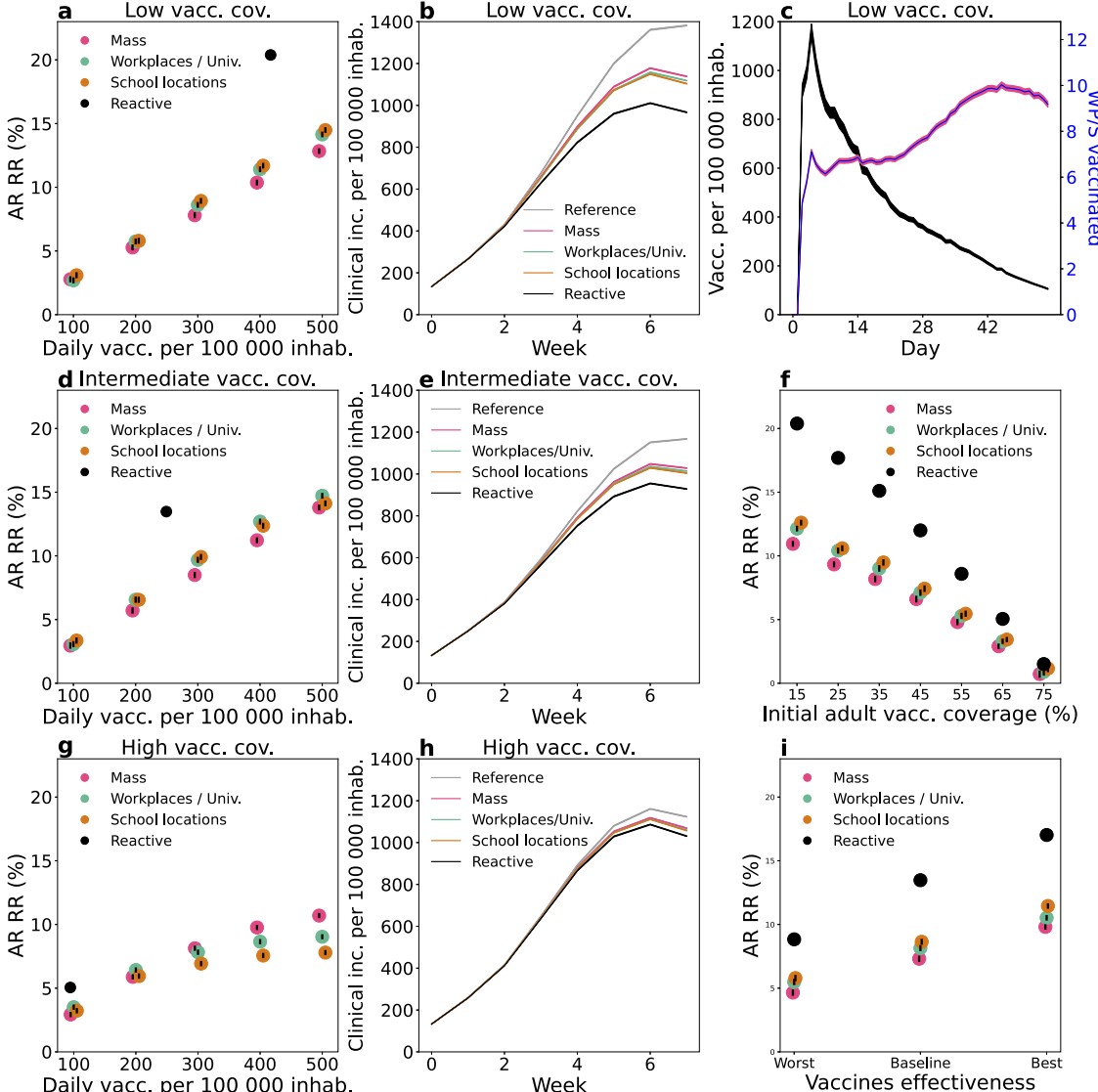

**Fig. 2 Comparison between vaccination strategies. a–h** Comparison between reactive and non-reactive vaccination strategies for the baseline scenario and different values of initial vaccination coverage. **a, d, g** Relative reduction (RR) in the attack rate (AR) over the first 2 months for all strategies as a function of the vaccination pace. RR is computed as $(AR_{ref} - AR)/AR_{ref}$ with $AR_{ref}$ being the AR of the reference scenario, where no vaccination campaign is conducted during the course of the simulation and vaccination coverage remains at its initial level. AR is computed from clinical cases. Three initial vaccination coverages are investigated: 15% of adults (low) (**a**); 40% of adults (intermediate) (**d**) and 65% of adults (high) (**g**). **b, e, h** Weekly incidence of clinical cases for 100,000 inhabitants for the first 8 weeks with different vaccination strategies. The non-reactive scenarios plotted are obtained with the same average daily vaccination pace as for reactive vaccination. Low, intermediate and high vaccination coverages are investigated in **b**, **e**, **h**, respectively. **c** Number of daily first-dose vaccinations, and number of workplaces/schools (WP/S in the plot) where vaccines are deployed for the same reactive scenario as in **a**, **b**—low vacc. cov., with 15% initial vaccine coverage. **f** AR RR for different initial vaccination coverages. The four strategies are compared at equal numbers of vaccine doses. The baseline epidemic scenario of panels **a–h** is defined by the following key parameters: $R = 1.6$; $VE_{S,1} = 48\%$, $VE_{SP,1} = 53\%$, $VE_{S,2} = 70\%$, $VE_{SP,2} = 73\%$; initial immunity 32%; initial incidence 160 clinical cases weekly per 100,000 inhabitants; 90% of 60+ vaccinated at the beginning. **i** AR RR for different vaccine effectiveness levels, assuming intermediate vaccination coverage (40% of adults) and all other parameters as in panels **a–h**. The baseline vaccine effectiveness values used in the other panels is compared with a worst and a best-case scenario, defined respectively by $VE_{S,1} = 30\%$, $VE_{SP,1} = 35\%$, $VE_{S,2} = 53\%$, $VE_{SP,2} = 60\%$, and by $VE_{S,1} = 65\%$, $VE_{SP,1} = 75\%$, $VE_{S,2} = 80\%$, $VE_{SP,2} = 95\%$. For each vaccine effectiveness scenario the four strategies are compared at equal numbers of vaccine doses. In panels **a**, **d**, **f**, **g**, **i**, data are means over 2000 independent stochastic realisations and error bars are derived from the standard error of the mean—these are smaller than the size of the dot in the majority of cases. In panels **b**, **c**, **e**, **h**, continuous lines are means over 2000 independent stochastic realisations and the shaded areas are the standard error of the mean (±2SEM)—not visible in panels **b**, **e**, **h**. The distribution of outcomes over all 2000 independent stochastic realisations is provided in Supplementary Fig. 3 comparing all vaccination strategies and considering the parameterisation of Fig. 2e as an example. The following abbreviations were used in the Fig.: vacc. for vaccines, inhab. for inhabitants, cov. for coverage, inc. for incidence, univ. for universities.

greater potential to transmit the infection. The reactive vaccination produced a stronger reduction in cases than the three other strategies in the 2-months period (black dot in Fig. 2a and black line in Fig. 2b). We found that 417 doses per 100,000 inhabitants

each day on average were used under the epidemic scenario considered here, yielding an attack-rate relative reduction of 20%.

In Fig. 2c we considered the same parametrisation as in Fig. 2a, b and we showed the number of first doses in time and the

number of places to vaccinate—as a proxy to the incurred logistics of a vaccine deployment. The number of daily inoculated doses was initially high, with almost 1200 doses per 100,000 inhabitants used in a day at the peak of vaccine demand, but declined rapidly afterwards down to 105 doses. The number of workplaces to vaccinate followed a different trend. It slowly increased to reach a peak and then declined. The breakdown in Supplementary Fig. 4 shows that schools and large workplaces were vaccinated at the very beginning. Thus a great number of vaccines were initially deployed in large settings, requiring many doses, while as the epidemic spread it reached a large number of small settings where only a few individuals could be vaccinated.

In Fig. 2d, e, we then considered an intermediate vaccination coverage at the beginning (40% of the [12,60] group and 90% of 60+, corresponding to ~45% of the whole population). Non-reactive strategies led to a relative reduction in the attack rate after 2 months close to that in the low initial coverage case, but the impact of reactive vaccination was reduced. Indeed, the proportion of unvaccinated people attending workplaces/schools that are targeted by vaccination is lower and fewer vaccine doses are administered, leading to a smaller impact at the population level (Fig. 2d). Still, reactive vaccination produced a 13% reduction in the attack rate using ~250 doses per 100,000 inhabitants each day on average, when the same reduction required ~400 doses per 100,000 inhabitants each day with non-reactive strategies. The impact of reactive vaccination finally became very small when initial vaccination coverage was high. Figure 2g, h shows a scenario where 65% of the [12,60] group and 90% of 60+ is vaccinated at the beginning, corresponding to ~60% of the whole population close to the coverage reached in Europe in the Fall 2021[1]. Only 94 daily vaccines per 100,000 inhabitants were used each day with a 5% reduction in the attack rate compared to a 3% reduction with non-reactive strategies for an equal number of doses. Non-reactive strategies with vaccination pace higher or equal to 300 doses per 100,000 inhabitants each day yielded a higher reduction in cases (~8% or higher).

The effect of the initial vaccination coverage on the impact of the different strategies is summarised in Fig. 2f. The relative reduction declined roughly linearly with the initial vaccination coverage. The reactive vaccination always outperformed non-reactive strategies at an equal number of doses. Nevertheless, the number of vaccinated people progressively decreased as initial vaccination coverage increased in the reactive vaccination approach, eventually reaching the point where it was less effective than non-reactive strategies with a large vaccination pace. In Fig. 2i we relaxed the baseline assumption on vaccine effectiveness and explored effectiveness parameters spanning the range of real-life estimates[17]. We found that lower vaccine effectiveness values led to a reduced effect of vaccination as expected. The difference between reactive and non-reactive strategies was also reduced.

In the Supplementary Information we compared reactive and non-reactive strategies under alternative epidemiological scenarios. In Supplementary Fig. 5 we assumed as a starting point the baseline scenario with intermediate vaccination coverage—i.e. the scenario in Fig. 2d, e with ~45% of the whole population vaccinated. We then varied key parameters, e.g. alternative values of transmission, incubation period, immunity level of the population, reduction in contacts due to social distancing, the time needed for the vaccine to become effective, compliance to vaccination and vaccine effect on the infection duration. An increase in the reproductive ratio, initial immunity and time between doses reduced the impact of the reactive vaccination. An increase in compliance to vaccination, instead, enhanced the impact of both reactive and non-reactive vaccination. Other parameters had a more limited role in strategies' effectiveness.

We then considered a scenario of a flare-up of cases, as it may be caused by a new variant of concern (VOC) spreading in the territory. In Supplementary Fig. 6a–c all parameters are as the baseline case of Fig. 2d, e, except for the initial incidence. The deployment of vaccines in this case was limited and slow. We then varied other parameters, i.e. the proportion of teleworking and time from building immunity following vaccination, finding that depending on their value the reactive strategy brought limited or no benefit with respect to non-reactive strategies, when the comparison was done at an equal number of doses (Supplementary Fig. 6d, e). Eventually, we tested the robustness of our results according to the selected health outcome, using hospitalisations, ICU admissions, ICU bed occupancy, deaths, life-years lost and quality-adjusted life-years lost, finding the same qualitative behaviour (Supplementary Fig. 7).

**Combined reactive and mass vaccination for managing sustained COVID-19 spread**. With the high availability of vaccine doses, reactive vaccination could be deployed on top of mass vaccination. We considered the baseline scenario with intermediate vaccination coverage defined in the previous section (i.e. ~45% of the whole population vaccinated, Fig. 2d, e) and compared mass and reactive vaccination simultaneously (combined strategy) with mass vaccination alone. We focused on the first 2 months since the implementation of the vaccination strategy. At an equal number of doses within the period, the combined strategy outperformed mass vaccination in reducing the attack rate. For instance, the relative reduction in the attack rate ranged from 10%, when ~360 daily doses per 100,000 inhabitants were deployed for mass vaccination, to 16%, when the same number of doses were used for reactive and mass vaccination combined (Fig. 3a).

We explored alternative scenarios where the number of vaccines used and places vaccinated were limited due to availability and logistic constraints. We assessed the effect of three parameters: (i) the maximum daily number of vaccines that can be allocated towards reactive vaccination (with caps going from 50 to 250 per 100,000 inhabitants, compared with unlimited vaccine availability assumed in the baseline scenario), (ii) the time from the detection of a case and the vaccine deployment (set to 2 days in the baseline scenario, and here explored between 1 and 4 days) and (iii) the number of detected cases that triggers vaccination in a place (from 2 to 5 cases, vs. the baseline value of 1). The number of first-dose vaccinations in time under the different caps is plotted in Fig. 3b. A cap on the number of doses limited the impact of the reactive strategy. Figure 3c shows that the attack-rate relative reduction dropped from 16 to 6% if only a maximum of 50 first doses per 100,000 inhabitants daily was used in reactive vaccination, reaching the levels of mass vaccination only. Doubling the time required to start reactive vaccination, from 2 days to 4 days, had a limited effect on the reduction of the AR (relative reduction reduced from 16 to 15%, Fig. 3d). Increasing the number of detected cases used to trigger vaccination to 2 (respectively, 5) reduced the relative reduction to 11% (respectively, 6%) (Fig. 3e).

We so far assumed that vaccine uptake was the same in mass and reactive vaccination. This assumption is likely conservative, in that individuals may be more inclined to accept vaccination when this is proposed in the context of reactive vaccination due to the higher perceived benefit of vaccination. In Fig. 3f we departed from the baseline assumption and considered a scenario where vaccine uptake with reactive vaccination climbed to 100%. Attack-rate relative reduction increased in this case from 16 to 22%, with a demand of ~480 daily doses per 100,000 inhabitants on average.

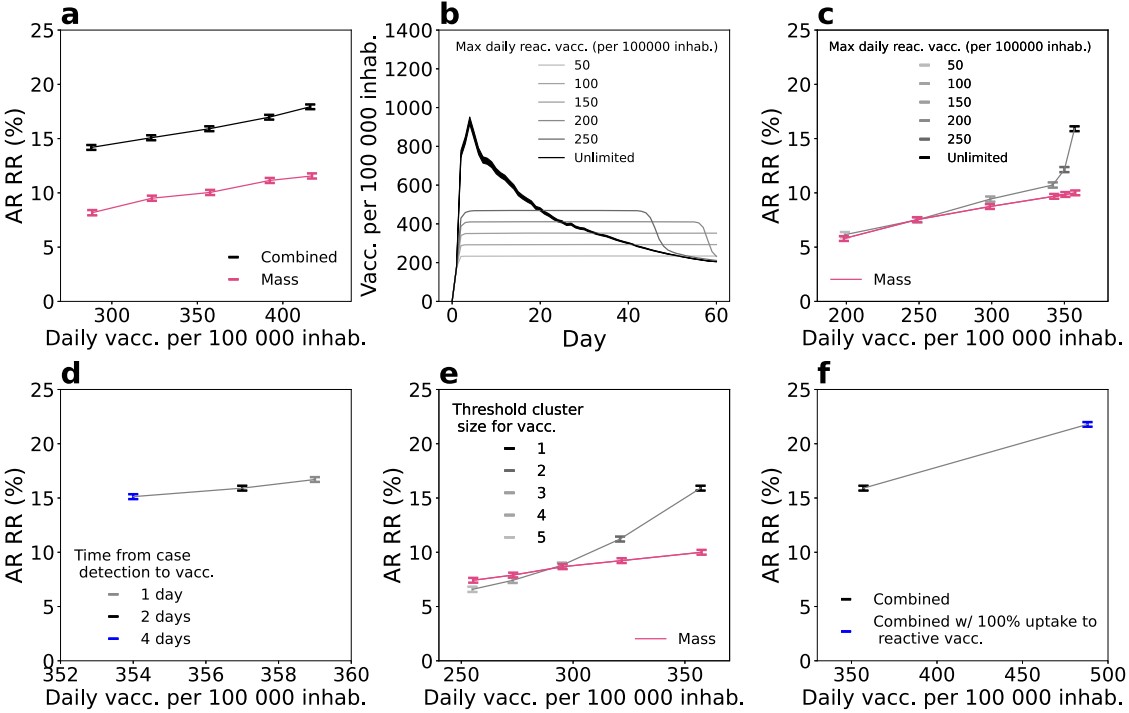

**Fig. 3 Combined reactive and mass vaccination for managing sustained COVID-19 spread. a** Relative reduction (RR) in the attack rate (AR) over the first 2 months for the combined strategy (mass and reactive) and the mass strategy with the same number of first-dose vaccinations as in the combined strategy during the period. RR is computed with respect to the reference scenario with initial vaccination only, as in Fig. 2. Combined strategy is obtained by running in parallel the mass strategy—from 50 to 250 daily vaccination rate per 100,000 inhabitants—and the reactive strategy. Number of doses displayed in the x-axis of the figure is the total number of doses used by the combined strategy, daily. Corresponding incidence curves are reported in Supplementary Fig. 8. **b** Number of first-dose vaccinations deployed each day for the combined strategy with different daily vaccines' capacity limits. **c**, **d**, **e** AR RR for the combined strategy as a function of the average daily number of first-dose vaccinations in the 2-months period. Symbols of different colours indicate: **c** different values of daily vaccines' capacity limit; **d** different time from case detection to vaccine deployment; **e** different threshold size for the cluster to trigger vaccination. In panel **c** and **e** the curve corresponding to mass vaccination only is also plotted for comparison. **f** Comparison between 100% and baseline vaccination uptake in case of reactive vaccination. Exception for the parameters indicated in the legend we assume in all panels baseline parameter values with intermediate vaccination coverage at the beginning—i.e. $R = 1.6$; $VE_{S,1} = 48\%$, $VE_{SP,1} = 53\%$, $VE_{S,2} = 70\%$, $VE_{SP,2} = 73\%$; initial immunity 32%; initial incidence 160 clinical cases weekly per 100,000 inhabitants; vaccinated at the beginning 90% and 40% for 60+ and <60, respectively. In panels **a**, **c**, **d**–**f** data are means over 2000 independent stochastic realisations and error bars are derived from the standard error of the mean. In panel **b** continuous lines are means over 2000 independent stochastic realisations and the shaded areas are the standard error of the mean (±2SEM)—only the standard error of the unlimited case is shown for clarity. The following abbreviations were used in the Fig.: vacc. for vaccines, inhab. for inhabitants, reac. for reactive.

**Combined reactive and mass vaccination for managing a COVID-19 flare-up.** We previously mentioned that in a scenario of a flare-up of cases reactive vaccination would bring limited benefit compared to other strategies (Supplementary Fig. 6). Here we analyse this scenario more in-depth assuming that reactive vaccination is combined with mass vaccination but triggers an increase in vaccine uptake and is associated with enhanced TTI, as may be the case in a realistic scenario of alert due to initial VOC detection. All other parameters were as in the baseline case, with intermediate vaccination coverage at the beginning (as in Fig. 2d, e).

We assumed mass vaccination with 150 first doses per day per 100,000 inhabitants was underway from the start, as well as baseline TTI. To start a simulation, three infectious individuals carrying a VOC were introduced in the population where the virus variant was not currently circulating. Upon detection of the first case, we assumed that TTI was enhanced, finding 70% of clinical cases, 30% of subclinical cases (i.e. ~45% of all cases) and three times more contacts outside the household with 100% compliance to isolation (Supplementary Table 4)—the scenario without TTI enhancement was also explored for comparison. As soon as the number of detected cases reached a predefined

threshold, reactive vaccination was started on top of the mass vaccination campaign. We assumed vaccine uptake increased to 100% for reactive vaccination but remained at its baseline value for mass vaccination.

In Fig. 4 we compare the combined scenario with mass vaccination alone at an equal number of doses and investigate starting reactive vaccination after 1, 5 or 10 detected cases. With reactive vaccination starting from the first detected case, the attack rate decreased by ~10%, compared with the mass scenario. However, the added value of reactive vaccination decreased if the start of the intervention was delayed. Without enhancement in TTI and increase in vaccine uptake, attack-rate values were much higher and the benefit of reactive vaccination over the mass vaccination was lower (~3%).

In Supplementary Fig. 10 we show different epidemic scenarios, testing different values for the transmissibility and vaccine effectiveness—including worst-case vaccine effectiveness, and R as high as 1.8—and found similar trends. Finally, we analysed the impact of vaccination on the flare-up extinction (Supplementary Fig. 11). With the parameterisation of Fig. 4a, the probabilities of extinction were 5.5% and 6% with mass and combined strategies, respectively. These values increased to

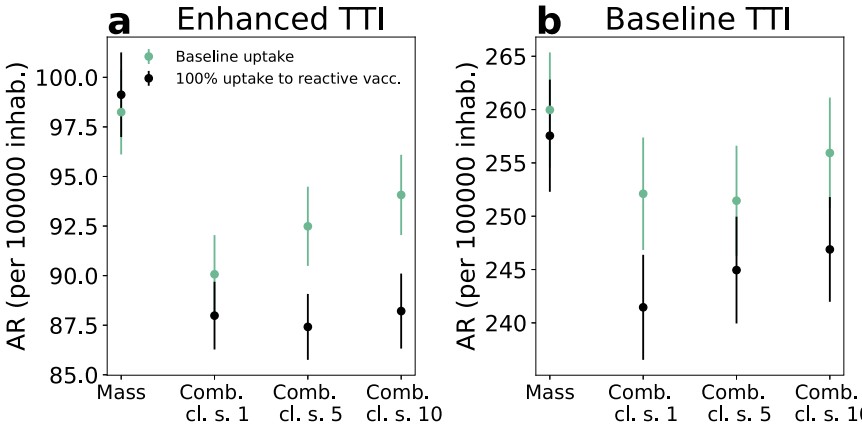

**Fig. 4 Combined reactive and mass vaccination for managing a COVID-19 flare-up. a, b** Attack rate (AR) per 100,000 inhabitants in the first 2 months for the enhanced (**a**) and baseline (**b**) TTI scenarios described in the main text. Four vaccination strategies are compared: mass only, combined where the reactive vaccination starts at the detection of 1, 5, 10 cases (Comb. cl. s. = 1, 5, 10 in the Fig.). For mass vaccination the number of first-dose vaccinations during the period is the same as in the comb. cl. s = 1 of the same scenario. Except when otherwise indicated parameters are the ones of the baseline epidemic scenario with intermediate vaccination coverage at the beginning—i.e. $R = 1.6$; $VE_{S,1} = 48\%$, $VE_{SP,1} = 53\%$, $VE_{S,2} = 70\%$, $VE_{SP,2} = 73\%$; initial immunity 32%; vaccinated at the beginning 90% and 40% for 60+ and <60, respectively. In both panels, data are means over 8000 independent stochastic realisations and error bars are the standard error of the mean (±2SEM). Corresponding incidence curves are reported in Supplementary Fig. 9. The following abbreviations were used in the Fig.: vacc. for vaccines, inhab. for inhabitants.

15% and 18% in a best-case scenario where vaccine protection occurred sooner after the first dose, vaccine efficacy was larger and TTI further strengthened.

## Discussion
The rapid rise of more transmissible SARS-CoV-2 variants has made the course of the COVID-19 pandemic unpredictable, posing a persistent public health threat that jeopardises the return to normal life[25,31–35]. More transmissible viruses call for vaccination of a larger portion of the population and increased accessibility and adaptation to a rapidly changing epidemic situation[2]. In this context, we analysed reactive vaccination of workplaces, universities and schools to assess its potential role in managing the epidemic.

The agent-based model used in this study accounted for the major factors affecting the effectiveness of reactive vaccination: disease natural history, vaccine characteristics, individual contact behaviour and logistical constraints. For a broad range of epidemic scenarios, reactive vaccination reduced the spread of COVID-19 more than non-reactive vaccination strategies—including untargeted mass vaccination—for an equal number of doses used over a period of 2 months. In addition, combining reactive and mass vaccination was more effective than mass vaccination alone. For instance, in a scenario of moderate/high incidence starting with 45% vaccination coverage, we found that the relative reduction in the attack rate over 2 months increased from 10 to 16% if 350 first vaccine doses per 100,000 habitants per day were used in a combined mass/reactive vaccination approach instead of mass only. However, the advantage of reactive vaccination was limited or nought with respect to non-reactive strategies under certain circumstances, as the number of doses administered with the reactive vaccination depended on the number and pace of occurrence and detection of COVID-19 cases. This occurred when vaccination coverage was already high at the beginning and only a few people could be vaccinated around detected cases, or in a flare-up scenario when only a few cases were detected. Non-reactive strategies could then be more effective provided the pace of vaccine administration was large enough. But even in these situations, adding reactive vaccination

to mass vaccination could become of interest again by triggering an increase in vaccine uptake, all the more when combined with enhanced TTI.

Reactive vaccination has been studied for smallpox, cholera and measles, among others[5–7,36,37]. Hotspot vaccination was found to help in cholera outbreak response in both modelling studies and outbreak investigation[37,38]. It may target geographic areas defined at spatial resolution as diverse as districts within a country, or neighbourhoods within a city, according to the situation. For Ebola and smallpox, ring vaccination was successfully adopted to accelerate epidemic containment[5–7]. These infections, though, have features making the approach a priori sensible: vaccine-induced immunity mounts rapidly compared to the incubation period and the mere absence of pre-symptomatic and asymptomatic transmission makes it possible to reach secondary cases before they start transmitting. Ring vaccination is also relevant when the vaccine has post-exposure effects[8]. Reactive vaccination in schools and university campuses has been implemented in the past to contain outbreaks of meningitis[39] and measles[40,41].

For COVID-19, the use of reactive vaccination has been reported in Ontario, the UK, Germany and France among others[2,42–47]. In these places, vaccines were directed to communities, neighbourhoods or building complexes with a large number of infections or presenting epidemic clusters or surge of cases due to virus variants. While the goal of these campaigns was to minimise the spread of the virus, it also addressed inequalities in access to healthcare and increased fairness, since a surge of cases may happen where people have difficulty in isolating due to poverty and house crowding[48]. In France, reactive vaccination was implemented to contain the emergence of variants of concern in the municipalities of Bordeaux, Strasbourg and Brest[45–47]. In the municipality of Strasbourg, vaccination slots dedicated to students were created following the identification of a Delta cluster in an art school[45]. Despite the interest in the strategy and its inclusion in the COVID-19 response plans, very limited work has been done so far to quantify its effectiveness[49,50]. A modelling study on ring vaccination suggested that the strategy could be valuable if the vaccine had post-exposure efficacy and a large proportion of contacts could be identified[50]. Still, post-exposure effects of the vaccine remain currently hypothetical[51], and it is likely that the vaccination of

the first ring of contacts alone would bring little benefit, if at all. We have here tested reactive vaccination of workplaces and schools, since focusing on these settings may be an efficient way to easily reach an extended group of contacts. Workplaces have been found to be an important setting for COVID-19 transmission, especially specific workplaces where conditions are more favourable for spreading[52,53]. The university setting also plays a central role in COVID-19 transmission, due to the high number of contacts among students, particularly if sharing common spaces in residence accommodations[54]. Model results show that reactive vaccination of these settings could have in many circumstances a stronger impact than simply reinforcing vaccination in these settings.

Importantly, the effectiveness of the reactive strategy depends on the epidemic context. We found that the higher the overall vaccination coverage, the less reactive vaccination would be of interest compared to non-reactive strategies if it did not increase vaccine uptake. For example, with >40% vaccination coverage among adults, the relative reduction in attack rates with reactive vaccination is smaller than with non-reactive alternatives provided large enough vaccination pace and no increase in vaccine uptake with reactive vaccination. Indeed, with large vaccination coverage, only a few individuals who have not been vaccinated before can be reactively proposed for vaccination, leading to few shots in case of vaccine hesitancy. Moreover, the detection of clusters is more difficult in a highly vaccinated population, where breakthrough infections in the vaccinated yield a large proportion of subclinical cases that are harder to detect.

If initial vaccination coverage is not too high, the feasibility and advantage of the inclusion of reactive vaccination imply a trade-off between epidemic intensity and logistical constraints. At a moderate/high incidence level, combining reactive and mass vaccination would substantially decrease the attack rate compared to mass vaccination for the same number of doses, but the large initial demand in vaccines may exceed the available stockpiles or capacity to deliver. The timely deployment of additional personnel in mobile vaccine units and the need to quickly inform the population by communication campaigns is indeed key to guarantee the success of the campaign. We explored with the model the key variables that would impact the strategy effectiveness. Delaying the deployment of vaccines in workplaces/schools upon the detection of a case (from 2 to 4 days on average) would not have a strong impact on its effectiveness. However, vaccines should be deployed at the detection of the first case to avoid substantially limiting the impact of the strategy—e.g. the relative reduction went from 16 to 6% when workplaces/schools were only vaccinated after the detection of 5 cases (Fig. 3e).

In the case of a COVID-19 flare-up the reactive strategy may bring an advantage if the reactive strategy starts early, is combined with increased TTI and triggers an increase in vaccine uptake. Starting early after the introduction of a first VOC case requires that tests for the detection of variants must be carried out regularly and with large coverage. Genomic surveillance has ramped up in many countries since the emergence of the Alpha variant in late 2020. For example, as of Fall 2021, nationwide surveys are conducted almost weekly in France to fully sequence the viral genome in randomly selected positive samples[55]. A proportion of positive tests are also routinely screened for key mutations to monitor the circulation of the main variants registered as VOC or VUI[55]. This surveillance protocol contributes to quickly identifying the presence of variants, but does not guarantee that interventions start with the few first cases, even more as the relaxation of social restrictions may lead to super spreading events. A strong intensification of TTI[23] must also be part of the wider response plan including reactive vaccination. Rapid and efficient TTI efficiently mitigates spread on its own, but it is also

instrumental to the success of reactive vaccination by triggering vaccination in households, workplaces and schools. Last, an increased level of vaccine uptake is essential for reactive vaccination to be of interest. Vaccination coverage remains highly heterogeneous worldwide and, as of Fall 2021, low in many countries of Eastern Europe and in many counties in the US[1,30]. Besides the individuals who oppose vaccination, a reactive strategy combined with the presence of a VOC may help increasing the acceptability of the vaccine by making it more accessible and anticipating an immediate benefit against the risk of infection. An increase in vaccine uptake was indeed observed in the context of a reactive vaccination campaign during the course of a measles outbreak[4]. Reactive vaccination could therefore be a means to improve access and acceptability in case of a flare-up.

The study is affected by several limitations. First, the synthetic population used in the study accounts for the repartition of contacts across workplaces, schools, households, etc., informed by contact surveys. However, number of contacts and risk of transmission may vary greatly according to the kind of occupation. The synthetic population accounts for this variability assuming that the average number of contacts from one workplace to another is gamma distributed[10], but no data were available to inform the model in this respect. Second, we model vaccination uptake according to age only, when it is determined by several socio-demographic factors. Clusters of vaccine-hesitant individuals may play an important role in the dynamics and facilitate the epidemic persistence in the population, as it is described for measles[56]. In those countries where vaccination coverage is high, heterogeneities in attitude toward vaccination may have an impact. Third, the agent-based model is calibrated from French socio-demographic data. The results of this study can be extended to countries with similar societal structure and contact patterns, e.g. other developed countries[57]. Still, COVID-19 transmission potential, level of disease-induced immunity, vaccination coverage, and extent of social restrictions vary substantially from one country to another. In addition, the waning of immunity since vaccination and recommendations for booster doses affect the level of protection of the population already vaccinated and consequently the impact of reactive vaccination. The large set of scenarios explored and reported in the Supplementary Information is intended to fully understand the interplay between epidemic spread and reactive vaccination and aid planning in case of future epidemic surges.

## Methods

**Synthetic population.** We used a synthetic population for a French municipality based on the National Institute of Statistics and Economic Studies (INSEE) censuses and French contact survey information[10,58]. This included the following input files: (i) a setting-specific, time-varying network of daily face-to-face contacts; (ii) the correspondence between individuals and their age, (iii) between individuals and the household they belong to, (iv) between individuals and their school, (v) and between individuals and their workplace. The synthetic population has an age pyramid, household composition, number of workplaces by size and number of schools by type, reproducing INSEE statistics. Daily face-to-face contacts among individuals are labelled according to the setting in which they occur (either household, workplace, school, community or transport) and they have assigned a daily frequency of activation, to explicitly model recurrent and sporadic contacts. We considered the municipality of Metz in the Grand Est region, which has 117,492 inhabitants, 131 schools (from kindergarten to University) and 2888 workplaces (Fig. 1a). A detailed description of how the population was generated is provided in[10]. Information about how to access population files is provided in the Data availability section.

**Overview of the model.** The model was written in C/C++, and is stochastic and discrete-time. It accounts for the following components: (i) teleworking and social distancing, (ii) COVID-19 transmission, accounting for the effect of the vaccine; (iii) test-trace-isolate; (iv) vaccine deployment. Model output included time series of incidence (clinical and subclinical cases), detailed information on infected cases (time of infection, age, vaccination status), vaccines administered according to the strategy, number of workplaces where vaccines are deployed. Different epidemic

scenarios were explored and compared. In the Supplementary Information we also analysed hospitalisation entries, deaths, ICU entries, life-year lost, quality-adjusted life-year, ICU bed occupancy. These quantities were computed by postprocessing output files containing detailed information on infected cases.

**Teleworking and social distancing**. Teleworking and other social restrictions may alter the repartition of contacts across settings and in turn the effectiveness of vaccination strategies[23]. We thus explicitly accounted for this ingredient in the model. Specifically, to model teleworking we assumed a proportion of individuals were absent from work, modelled by erasing working contacts and transport contacts of these individuals. To account for the reduction in social encounters due to the closure of restaurants and other leisure activities we removed a proportion of contacts from the community layer. In Western countries, the level of restrictions varied greatly both by country and in the time since vaccines were first deployed at the beginning of 2021. We set the contact reduction in the community to 5% and the teleworking to 10%. These were close to the reduction values reported by google mobility reports for France during Autumn 2021[59], and fell within the range of European countries' estimates. Note that levels of teleworking ~10% for European countries were reported also by other sources[60]. Scenarios with different levels of contact reduction were compared in the Supplementary Information. Telework and social distancing were implemented at the beginning of the simulation and remained constant for the duration of the simulation. Importantly, the reproductive ratio was set to the desired value, independently by the level of contact reduction, as described in the Supplementary Information.

**COVID-19 transmission model**. We used an extension of the transmission model in ref. [10] (see Fig. 1c). This accounted for heterogeneous susceptibility and severity across age groups[61,62], the presence of an exposed and a pre-symptomatic stage[9] and two different levels of infection outcome—subclinical, corresponding to asymptomatic or paucisymptomatic infection and clinical, corresponding to moderate to critical infection[61,63]. Precisely, susceptible individuals, if in contact with infectious ones, could get infected and enter the exposed compartment ($E$). After an average latency period $\epsilon^{-1}$ they became infectious, developing a subclinical infection ($I_{sc}$) with age-dependent probability $p_{sc}^A$ and a clinical infection ($I_c$) otherwise. From $E$, before entering either $I_{sc}$ or $I_c$, individuals entered first a prodromal phase (either $I_{p,sc}$ or $I_{p,c}$, respectively), that lasted on average $\mu_p^{-1}$ days. Compared to $I_{p,c}$ and $I_c$ individuals, individuals in the $I_{p,sc}$ and $I_{sc}$ compartments had reduced transmissibility rescaled by a factor $\beta_I$. With rate $\mu$ infected individuals became recovered. Age-dependent susceptibility and age-dependant probability of clinical symptoms were parametrised from[61]. In addition, transmission depended on setting as in[10]. We assumed that the time spent in the $E$, $I_{p,sc}$ and $I_{p,c}$ was Erlang distributed with shape 2, and rate $2\epsilon$ for $E$, and $2\mu_p$ for $I_{p,sc}$ and $I_{p,c}$. Time spent in $I_{sc}$ and $I_c$ was exponentially distributed. Parameters and their values are summarised in Supplementary Table 1.

We modelled vaccination with a leaky vaccine, partially reducing both the risk of infection (i.e. reduction in susceptibility, $VE_S$) and infection-confirmed symptomatic illness ($VE_{SP}$)[15]. The level of protection increased progressively after the inoculation of the first dose. In our model we did not explicitly account for the two-dose administration, but we accounted for two levels of protection—e.g. a first one approximately in between the two doses and a second one after the second dose. Vaccine efficacy was zero immediately after inoculation, mounting then to an intermediate level ($VE_{S,1}$ and $VE_{SP,1}$) and a maximum level later ($VE_{S,2}$ and $VE_{SP,2}$). This is represented through the compartmental model in Fig. 1c. Upon administering the first dose, $S$ individuals became, $S^{V,0}$, i.e. individuals that are vaccinated, but have no vaccine protection. If they did not become infected, they entered stage $S^{V,1}$, where they were partially protected, then stage $S^{V,2}$ where vaccine protection was maximum. Time spent in $S^{V,0}$ and $S^{V,1}$ was Erlang distributed with shape 2 and rate $2/\tau_0$ and $2/\tau_1$ for $S^{V,0}$ and $S^{V,1}$, respectively. $S^{V,1}$ and $S^{V,2}$ individuals had reduced probability of getting infected by a factor $r_{S,1} = (1 - VE_{S,1})$ and $r_{S,2} = (1 - VE_{S,2})$, respectively. In case of infection, $S^{V,2}$ individuals progressed first to exposed vaccinated ($E^V$), then to either preclinical or pre-subclinical vaccinated ($I_{p,c}^V$ or $I_{p,sc}^V$) that were followed by clinical and subclinical vaccinated, respectively ($I_c^V$ or $I_{sc}^V$). Probability of becoming $I_{p,c}^V$ from $E^V$ was reduced of a factor $r_{c,2} = (1 - VE_{SP,2})(1 - VE_{S,2})^{-1}$. For the $S^{V,1}$ individuals that get infected we assumed a polarised vaccine effect, i.e. they can enter either in $E^V$, with probability $p_V$, or in $E$ (Fig. 1c). The value of $p_V$ was set based on $VE_{SP,1}$ through the relation $(1 - VE_{SP,1}) = (1 - VE_{S,1})(p_V r_{c,2} + (1 - p_V))$. We assumed no reduction in infectiousness for vaccinated individuals. However, we accounted for a 25% reduction in the duration of the infectious period as reported in refs. [64,65].

Under the assumption that no serological/virological/antigenic test is done before vaccine administration, the vaccine was administered to all individuals, except for clinical cases who showed clear signs of the disease or individuals that were detected as infected by the TTI in place. In our model a vaccine administered to infected or recovered individuals had no effect.

In the baseline scenario we parametrised $VE_{SP,1}$, $VE_{SP,2}$, $VE_{S,1}$ and $VE_{S,2}$ by taking values in the middle of estimates reported in the systematic review by Higdon and collaborators[17] for the Comirnaty vaccine and the Delta variant[66]. Chosen values of $VE_{SP,2}$, and $VE_{S,2}$ are also comparable with the effectiveness

estimates reported in a meta-analysis for the Delta variant, complete vaccination, all vaccines combined[18]. We also tested values on the upper and lower extremes of the range of estimates of[17]. Parameters are listed in Supplementary Table 2.

**Test-trace-isolate**. We model a baseline TTI accounting for case detection, household isolation and manual contact tracing. Fifty percent of individuals with clinical symptoms were assumed to get tested and to isolate if positive. We assumed an exponentially distributed delay from symptoms onset to case detection and its isolation with 3.6 days on average. Once a case was detected, his/her household members isolated with probability $p_{ct,HH}$, while other contacts isolated with probabilities $p_{ct,A}$ and $p_{ct,Oth}$, for acquaintances and sporadic contacts, respectively. In addition to the detection of clinical cases, we assumed that a proportion of subclinical cases were also identified (10%). Isolated individuals resumed normal daily life after 10 days unless they still had clinical symptoms after the time had passed. They could, however, decide to drop out from isolation each day with a probability of 13% if they did not have symptoms[67].

In the scenario of virus re-introduction we considered enhanced TTI, corresponding to a situation of case investigation, screening campaign and sensibilisation (prompting higher compliance to isolation). We assumed a higher detection of clinical and subclinical cases (70% and 30%, respectively), perfect compliance to isolation by the index case and household members and a three-fold increase in contacts identified outside the household.

A step-by-step description of contact tracing is provided in the Supplementary Information. Parameters for baseline TTI are provided in Supplementary Table 3, while parameters for enhanced TTI are provided in Supplementary Table 4.

**Vaccination strategies**. A vaccine opinion (willingness or not to vaccinate) was stochastically assigned to each individual at the beginning of the simulation depending on age (below/above 65 years old). The opinion did not change during the simulation. In some scenarios we assumed that all individuals were willing to accept the vaccine in case of reactive vaccination, while maintaining the opinion originally assigned to them when the vaccine was proposed in the context of non-reactive vaccination. Only individuals above a threshold age, $a_{th,V} = 12$ years old, were vaccinated. We assumed that a certain fraction of individuals were vaccinated at the beginning of the simulation according to the age group ([12,60], 60+). We compared the following vaccination strategies:

*Mass.* $V_{daily}$ randomly selected individuals were vaccinated each day until a $V_{tot}$ limit was reached.

*Workplaces/universities.* Random workplaces/universities were selected each day. All individuals belonging to the place, willing to be vaccinated, and not isolated at home that day were vaccinated. Individuals in workplaces/universities were vaccinated each day until the daily limit, $V_{daily}$, was reached. No more than $V_{tot}$ individuals were vaccinated during the course of the simulation. We assumed that only workplaces with $size_{th} = 20$ employees or larger implemented vaccination.

*School location.* Random schools, other than universities, were selected each day and a vaccination campaign was conducted in the places open to all household members of school students. All household members willing to be vaccinated, above the threshold age and not isolated at home that day were vaccinated. No more than $V_{daily}$ individuals were vaccinated each day and no more than $V_{tot}$ individuals were vaccinated during the course of the simulation.

*Reactive.* When a case was detected, vaccination was done in her/his household with rate $r_V$. When a cluster—i.e. at least $n_{cl}$ cases detected within a time window of length $T_{cl}$—was detected in a workplace/school, vaccination was done in that place with rate $r_V$. In the baseline scenario, we assumed vaccination in the workplace/school was triggered by one single infected individual ($n_{cl} = 1$). In both households and workplaces/schools, all individuals belonging to the place above the threshold age and willing to be vaccinated were vaccinated. Individuals that were already detected and isolated at home were not vaccinated. No more than $V_{daily}$ individuals were vaccinated each day and no more than $V_{tot}$ individuals were vaccinated during the course of the simulation. In the baseline scenario these quantities were unlimited, i.e. all individuals to be vaccinated in the context of reactive vaccination were vaccinated.

Parameters and their values are summarised in Supplementary Table 5.

**Reporting summary**. Further information on research design is available in the Nature Research Reporting Summary linked to this article.

## Data availability
The synthetic population used in the analysis is available on zenodo[68].

## Code availability
We provide all C/C++ code files of the model on zenodo[68].

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

## Acknowledgements

We acknowledge financial support from Haute Autorité de Santé; the ANR and Fondation de France through the project NoCOV (00105995); the Municipality of Paris (https://www.paris.fr/) through the programme Emergence(s); EU H2020 grants MOOD (H2020-874850; paper MOOD 035) and RECOVER (H2020-101003589) (the contents of this publication do not necessarily reflect the views of the European Commission); the ANRS through the project EMERGEN (ANRS0151); and the Institut des Sciences du Calcul et de la Donnée.

## Author contributions

V.C., P.Y.B and C.P. designed the analysis. P.Y.B. and C.P. developed the main methodology. B.F., R.A. and J.R. performed the analysis. D.L.B., C.T.K., S.C. and L.Z. critically commented on assumptions and model structure. P.Y.B. and C.P. drafted the manuscript. All authors discussed the results, edited and approved the contents of the manuscript.

## Competing interests

The authors declare no competing interests.

## Additional information

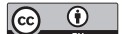

