## [Peer Review File · Nature Communications]

Agent-based modelling of reactive vaccination of workplaces and schools against COVID-19REVIEWER COMMENTS

Reviewer #1 (Remarks to the Author):

Thank you for the opportunity to review this work on targeted vaccination, which I read with interest. The paper investigates reactive vaccination for COVID-19, using a detailed model of COVID-19 transmission linked with a mitigation response system combining various types of vaccination and targeted NPIs. In general, I think the model incorporates many of the features necessary to act as an effective guide for policy recommendations (which, in my interpretation, is the intended application of the model).

The work investigates recommendations around targeted, reactive vaccination strategies in which transmission environments are prioritised for vaccination after the discovery of new cases. As the authors note, such policies have been effective for pathogens such as Ebola. The authors claim that such policies would be more effective than mass vaccination or statically-targeted vaccination in mitigating COVID-19. While it does seem like a sensible investigation, I do not believe the argument is as compelling as it needs to be to support the conclusion. Given the current scope of the model's assumptions and the parameter space explored, it is not clear that the policy recommendation is robust.

Specifically, the following assumptions need to be relaxed and investigated in order to make the argument robust, and to establish the limitations of the policy: (1) incubation period distribution, (2) distribution of time from inoculation to onset of protection, (3) comparison with targeted (static) strategies in the sensitivity analysis.

Details:

(1) Currently, the incubation period is modelled as the combination of two state dwell times, each with exponentially distributed duration. This produces gamma distributed incubation periods with a good match to the mean (5.2 days), but with a fairly high density of shorter-than-average delays from infection to symptom onset (note that the incubation period used in the model is based on a very early report that used data from only 10 cases [Li et al, 2020], better estimates are now available e.g., Lauer et al. DOI: 10.7326/M20-0504). This consideration may be important with respect to the detection process (cases may be detected too soon). Furthermore, in my reading of the model I understand that only the second of these incubation states is infectious, which gives exponentially distributed pre-symptomatic infectious periods that average 2.3 days. This assumption should be relaxed, as I do not believe this will reproduce the very large fraction of pre-symptomatic transmissions observed for COVID-19 (this can be on the order of 40% - see e.g., Ferretti et al. 2020 "The timing of COVID-19 transmission"). Altogether, underestimating pre-symptomatic transmission and over-estimating the rate of case detection could lead to overestimation in the relative effectiveness of reactive strategies.

(2) In the current version, two sets of parameters have been tested for the rates of onset of partial and maximum protection: $v_1 = v_2 = 1/1\text{wk}$ and $v_1 = v_2 = 1/2\text{wk}$ (explored in the supplement). There are two aspects of this that are currently inadequate: i) the exponentially distributed times between inoculation and protection (giving a large fraction of the population very short delays to protection), and ii) the distribution averages (2 or 4 weeks to maximum immunity), which are substantially shorter than what is typically achieved in practice.

Current recommendations range between 4 and 12 weeks for the delay between first and second doses, with about 14-21 days between inoculation and onset of protection from the first dose (See Harris et al. 2021 DOI: 10.1056/NEJMc2107717). Care should be taken to examine realistic time scales for these parameters, because the policy recommendations supported by the results will be sensitive to them.

(3) Regarding these recommendations, when benchmarking the reactive strategies in the sensitivity analysis, comparison should be made between reactive and targeted strategies (i.e., those that specifically target high-transmission environments, regardless of case detection as shown in Fig 2A – green and orange bars). This will ensure a fair comparison of the reactive

strategy to alternative static strategies.

Logically, for a long enough duration between inoculation and protection, targeted vaccination (or even random vaccination) should be more beneficial than reactive vaccination. This is because the reactive strategy will be too slow and will produce redundant vaccination of individuals who are already infected or recovered. This threshold should be reported so that the limitations of the strategy are clear.

In Figure 2A, 255 daily vaccinations gives AR RR of about 14% for school targeting, while in Figure S4E, reactive vaccination with a delay of $v1 = v2 = 1/14d$ gives AR RR of 17-18%. Therefore, I believe that even a modest relaxation of assumptions as pointed out in (1) and (2) above, towards increased realism, may weaken or reverse the argument favoring reactive vaccination policies.

Overall, the modelling approach and framework is thorough and well-presented. If the comments above are addressed, I would support publication of the paper in revised form. If addressing the points above weakens or reverses the conclusion with respect to the proposed advantages of reactive vaccination, I would still support publication as long as the result is robust.

Regards,

Cameron Zachreson

Reviewer #2 (Remarks to the Author):

The authors here tackle an important problem in COVID-19 epidemiology which is about the efficient use of vaccination to control outbreaks. They use at its basis a detailed multi-layer network model based on known population characteristics based on a single region (the city of Metz) where the network is dynamic and in simulation, vaccines can be dynamically allocated. Vaccine uptake rates are age dependent but do not appear to otherwise vary.

The model itself has been published and is well established. While the work overall is extensive and informative, I do have concerns about the extent to which the conclusions of the paper are relevant to future of COVID control under current conditions as the assumptions, while undoubtedly merited when this work was originally done, do seem to miss some key points about the delta variant in particular. While the basic recommendations themselves about the value of targeted vaccination as a general principle, and the importance of picking up potential clusters of infection (in scenarios where transmission risk is either high and/or important) are well established, even without a modelling analysis, the specifics of the modelling analysis are likely to be more dependent on the characteristics of the virus, as modelled. Thus compromises what modelling can add to the situation in terms of evaluation of the specifics of when, how quickly and with what benefit. Here however, some of the limitations show themselves. As we all know the COVID-19 situation evolves rapidly.

Here for example, current evidence suggests that the delta variant is substantially more transmissible than previous dominant types (and so is the $R = 1.2$ valid under most future conditions where we are reliant on vaccination to prevent deaths rather than further extensive restrictions), causes different symptoms, and with reduced vaccine efficacy compared to previous variants ([https://www.thelancet.com/journals/lancet/article/PIIS0140-6736\(21\)01290-3/fulltext](https://www.thelancet.com/journals/lancet/article/PIIS0140-6736(21)01290-3/fulltext)); recent evidence also suggests that vaccinated individuals are as likely to reach high viral loads in throat if infected, as unvaccinated individuals (measured through C_t counts - see for example here <https://www.medrxiv.org/content/10.1101/2021.02.16.21251535v3.full.pdf>) and therefore transmit substantially to transmission, but also clear it more efficiently.

The assumption that all symptomatic individuals get picked up is quite optimistic: this is evident in the UK data which I am more familiar with due to the high number of individuals who are tested and are positive when entering hospital - i.e. not nosocomial infections but likely individuals who went to hospital because they were seriously ill and only tested positive then. Thus there are likely to be many people who have relatively mild infections, who never get tested. Comparisons to

random surveillance data suggests that the number may be as low as one in four ()

Reduction of transmission in the models, is based on google mobility data - it is unclear how well correlated these two factors are - certainly in the UK, mobility has been changing at rates that seem to have little correlation to either infection rates or estimates of R, and are quite different from contact rates estimated by the "COMIX" survey (<https://bmcmmedicine.biomedcentral.com/articles/10.1186/s12916-020-01597-8>) and its the latter which seems to correlate well with changes in R.

I would stress that the analysis itself seems solid and is based on an established model. However, because of the ever changing COVID situation, I do not see it as being of important general interest - the general aspects are good but not novel while the individual analyses have novelty but aren't currently relevant.

Reviewer #3 (Remarks to the Author):

The authors built an individual-based model to explore the impact of different vaccination strategies, especially the reactive vaccination of workplaces and schools. I believe this is a novel study that is almost ready to publish. Before that, some issues need to be addresses:

1. In model shown in Figure 3C, table S3 and table S4, the probability of detecting a subclinical case is set as .1 and .5 for both unvaccinated infectious and vaccinated infectious. I assume for vaccinated subclinical cases, the probability of detecting is much smaller than the clinical cases. I wonder, whether the heterogeneity of probability in unvaccinated and vaccinated subclinical cases makes a huge difference? It seems that the Delta Variant breakthrough infections spreading is common (even if with a shorter infectious period). If the vaccinated subclinical transmission is common, I imagine the surveillance will be very effective to curb the Delta spreading.

2. I had a hard time understanding Figure 2.

(a) In panel B legend, is "Random" the same as "Mass" in other panels?

(b) In line 133-134, "Figure 2B" should be consistent with "Panel A".

(c) The description for Figure 2 panel A-B is a bit back-and-forth: the authors first briefly described panel A and panel B then only talked about panel A. Can the authors talk more about panel B?

Answer to the comments of Reviewer 1

“Thank you for the opportunity to review this work on targeted vaccination, which I read with interest. The paper investigates reactive vaccination for COVID-19, using a detailed model of COVID-19 transmission linked with a mitigation response system combining various types of vaccination and targeted NPIs. In general, I think the model incorporates many of the features necessary to act as an effective guide for policy recommendations (which, in my interpretation, is the intended application of the model).”

The work investigates recommendations around targeted, reactive vaccination strategies in which transmission environments are prioritised for vaccination after the discovery of new cases. As the authors note, such policies have been effective for pathogens such as Ebola. The authors claim that such policies would be more effective than mass vaccination or statically-targeted vaccination in mitigating COVID-19. While it does seem like a sensible investigation, I do not believe the argument is as compelling as it needs to be to support the conclusion. Given the current scope of the model's assumptions and the parameter space explored, it is not clear that the policy recommendation is robust.

Specifically, the following assumptions need to be relaxed and investigated in order to make the argument robust, and to establish the limitations of the policy: (1) incubation period distribution, (2) distribution of time from inoculation to onset of protection, (3) comparison with targeted (static) strategies in the sensitivity analysis.”

We thank the Reviewer for recognising the interest of the work and its potential value as a guide for policy recommendation. We have carefully addressed the three points raised, modifying the computational model to make it more realistic, revising the parameterization of the baseline scenarios analysed, and making the sensitivity analysis more extensive.

Please note that, as detailed in the response to the Editor and Reviewer 2, we have paid attention to updating the work according to the evolving COVID-19 epidemic. In particular: 1) Reviewer 2 remarked that the Delta variant, now dominant in many part of the world, differs from the wildtype in many aspects, we have thus parameterised the model based on evidence for the Delta variant; 2) the epidemic situation and the vaccination campaign has evolved since the time the original manuscript was prepared we have thus changed epidemic and vaccination parameters to better reflect the current situation in many Western countries.

“Details:

(1) Currently, the incubation period is modelled as the combination of two state dwell times, each with exponentially distributed duration. This produces gamma distributed incubation periods with a good match to the mean (5.2 days), but with a fairly high density of shorter-than-average delays from infection to symptom onset (note that the incubation period used in the model is based on a very early report that used data from only 10 cases [Li et al, 2020], better estimates are now available e.g., Lauer et al. DOI: 10.7326/M20-0504). This consideration may be important with respect to the detection process (cases may be detected too soon). Furthermore, in my reading of the model I understand that only the second of these incubation states is infectious, which gives exponentially distributed pre-symptomatic infectious periods that average 2.3 days. This assumption should be relaxed, as I do not believe this will reproduce the very large fraction of pre-symptomatic transmissions observed for COVID-19 (this can be on the order of 40% - see e.g., Ferretti et al. 2020 "The timing of COVID-19 transmission"). Altogether, underestimating pre-symptomatic transmission and over-estimating the rate of case detection could lead to overestimation in the relative effectiveness of reactive strategies.”

Dwell time distributions: We acknowledge that our choice of distributions could lead to biased results if it leads to identifying cases too early. Therefore, we changed the model to account for a more realistic distribution of the latent and the pre-symptomatic period. More specifically, we used the “gamma trick” and split in two each of the compartments E, $I_{p,c}$, $I_{p,sc}$ (see Figure 1 in the main paper) (i.e E as $E1 \rightarrow E2$) while doubling the rates so that the average remain the same.

As the reviewer mentioned, the concatenation of several exponential distributions results in a Gamma distribution (Wearing et al PLOS Med 2005). The same strategy was adopted for the time from vaccine inoculation to protection to address the Reviewer’s point below, splitting the sojourn time in each vaccination stage (compartments $S_{v,0}$ and $S_{v,1}$ in Figure 1 of the main paper).

All results presented in the main text and Supplementary Information are now obtained with the revised version of the model. Also, we have updated the text - Methods section, Supplementary Information - to describe this modelling feature.

Results presented in the current version of the paper differ from the original version in different aspects. Beside the model assumption above mentioned, the parameters were also changed as described throughout this rebuttal letter. To clearly understand how changing the dwell time distribution impacted the result we provide here the direct comparison between the model with and without the split by keeping all other parameters unchanged.

The Figure below shows: A) the attack-rate relative reduction with respect to the reference scenario (i.e. where no vaccination campaign is conducted during the course of the simulation and vaccination coverage remains at its initial level) for mass and reactive vaccination; B) the incidence for mass and reactive vaccination. In both panels we compare the cases in which: 1) no split is implemented in the compartments (original version of the model); 2) the split is implemented only in the compartments describing the disease natural history (i.e. E, $I_{p,c}$, $I_{p,sc}$); 3) the split is implemented only in the compartment related to immunity following vaccination (i.e. $S_{v,0}$ and $S_{v,1}$ stages); 4) a combination of point 2) and 4), i.e. the split is implemented in E, $I_{p,c}$, $I_{p,sc}$, $S_{v,0}$ and $S_{v,1}$, as in the current version of the model. We obtain that, when compartments are splitted the impact of vaccination is reduced for both mass and reactive vaccination, confirming the hypothesis of the Reviewer. The effect is stronger when the split is done in the $S_{v,0}$ and $S_{v,1}$ compartments.

Figure. **A** Relative reduction (RR) in the attack rate (AR) for mass and reactive vaccination with respect to the reference scenario, where vaccination is implemented only at the beginning. **B** Incidence for both mass and reactive vaccination. In both panels we compare four model assumptions, where either one or two compartments are considered for the exposed and pre-symptomatic stages, E , $I_{p,c}$, $I_{p,sc}$, ($e1$ and $e2$ in the figure, respectively), and either one or two compartments are considered for the vaccinations stages, i.e. $S_{v,0}$ and $S_{v,1}$, ($v1$ and $v2$ in the figure, respectively). We consider here the scenario with moderate/high incidence and intermediate initial vaccination coverage, i.e. the same parameterization used in Figure 2 E of the main paper. For each model implementation, mass and reactive vaccination are compared at an equal number of doses.

Incubation period and pre-symptomatic transmission: We acknowledge that the incubation period was parameterized with evidence from an early study. Also, Reviewer 2 pointed out the importance of accounting for the specificity of the Delta variant. Therefore, we set the incubation period to 5.8 days based on a recently published analysis of a Delta outbreak in China (Kang et al medRxiv <https://doi.org/10.1101/2021.08.12.21261991>). In the same study authors provided evidence that the proportion of presymptomatic transmission was around 74% for close contacts in a context with case finding and quarantining. As noted by the reviewer, choosing a low proportion of pre-symptomatic transmission would presumably increase the effect of reactive vaccination. Therefore, we calibrated the value of the average durations of the latent period and prodromic phase to be on par with the 74% estimate, leading to a latency period of 3.7 days and a pre-symptomatic transmission duration of 2.1 days. The fraction of pre-symptomatic transmission was computed by analysing the model output with the list of all transmission events, with information of the infection status of the infector. This is now specified in the Supplementary Information.

All the results presented in the paper (main text and Supplementary Information) are now obtained with the new parameterization.

In the Supplementary Information we now include a sensitivity analysis on the value of the incubation period, exploring 5.1 as reported by Lauer et al. DOI: 10.7326/M20-0504, mentioned by the Reviewer, and 6.3, obtained in a recent systematic review and meta-analysis (Xin et al Clinical Infectious Diseases 2021 <https://academic.oup.com/cid/advance->

[article/doi/10.1093/cid/ciab501/6297425](https://doi.org/10.1093/cid/ciab501/6297425)). We find that results are quite robust to the choice of this parameter within this range.

“(2) In the current version, two sets of parameters have been tested for the rates of onset of partial and maximum protection: $v_1 = v_2 = 1/1\text{wk}$ and $v_1 = v_2 = 1/2\text{wk}$ (explored in the supplement). There are two aspects of this that are currently inadequate: i) the exponentially distributed times between inoculation and protection (giving a large fraction of the population very short delays to protection), and ii) the distribution averages (2 or 4 weeks to maximum immunity), which are substantially shorter than what is typically achieved in practice.

Current recommendations range between 4 and 12 weeks for the delay between first and second doses, with about 14-21 days between inoculation and onset of protection from the first dose (See Harris et al. 2021 DOI: 10.1056/NEJMc2107717). Care should be taken to examine realistic time scales for these parameters, because the policy recommendations supported by the results will be sensitive to them.”

Dwell time distributions in the vaccine-protection stages: To better model the delay from inoculation to protection we have modified the model to use an Erlang distributed dwell time in each vaccination stage (compartments $S_{v,0}$ and $S_{v,1}$ in Figure 1 of the main paper). This is discussed already in the reply to the point (1) above. The new version of the model has been used to generate all results presented in the manuscript.

Duration of vaccine-protection stages: We have now made it clear that we consider a vaccination strategy based on the Cominarty vaccine, that is highly available, can be deployed at a three-weeks interval between the two doses and is highly effective, all these factors making it the most suitable for reactive vaccination.

The choice of using $v_1=v_2=1/1\text{wk}$ was motivated by the results of the Cominarty phase III trial that showed evidence that approximately 10 days after the first dose the cumulative incidence in the vaccine and Placebo groups diverge. This was also supported by a JCVI statement of last January, showing that vaccine efficacy is 90% 2 weeks after the first dose (<https://www.gov.uk/government/publications/prioritising-the-first-covid-19-vaccine-dose-jcvi-statement/optimising-the-covid-19-vaccination-programme-for-maximum-short-term-impact>).

We appreciate that the time of onset of protection after inoculation and the efficacy after one and two doses in vaccine protection may depend on the circulating variant. To follow the advice of Reviewer 2, we re-parameterized VE parameters based on recent studies on VE against Delta. In many cases VE was assessed 2 weeks after each dose inoculation, i.e. an individual was defined as partially vaccinated starting from 2 weeks after dose 1, and fully vaccinated starting from 2 weeks after dose 2 (e.g. Nasreen et al <https://doi.org/10.1101/2021.06.28.21259420>). We thus made the conservative hypothesis that protection is reached two weeks after each dose injection. The Cominarty vaccine was originally authorised at a dosing interval of 3 weeks. However, recommendations concerning the dosing interval varied in time and from country to country, in relation to several factors, including

vaccine availability, the epidemiological situation and the circulation of the delta variant. In particular, a dosing interval of 3 weeks was recommended by, e.g. Canada, France and United States to contain the epidemic surge due to the Delta variant. We presume that in a context of reactive vaccination, building protection as fast as possible with short intervals between doses would be preferred, all the more than vaccine availability is not an issue anymore. Therefore, we made the baseline hypothesis that VE is 0 for the first 2 weeks on average, at an intermediate level for the following period of 3 weeks on average, and at maximum level after. In the Supplementary Information we analyse the impact of the delay between doses, exploring values up to 8 weeks. We found that, in a scenario of low initial incidence (flare-up scenario) the reactive vaccination leads to a reduction comparable or even lower than non-reactive strategies for a long interval between vaccine doses.

Note that we have also updated the vaccine effectiveness parameters to account for the reduced effectiveness against Delta, especially after one dose. Estimates of VE for the Delta variant are real life estimates, obtained with different study designs and potentially subject to bias: these are affected by the complex interplay among delta variant emergence, waning of immunity and differential impact by age. We relied on the results of the systematic review by Higdon et al (medRxiv 2021, <https://doi.org/10.1101/2021.09.17.21263549>) which reports VE_S,1 being between 30% and 65 %, VE_SP,1 between 35% and 75%, VE_S,2 between 60% and 80%, and VE_SP,2 between 55% and 90 %. We set our baseline scenario to the middle of these ranges with VE_S,1 = 48% ; VE_SP,1 = 55% ; VE_S,2 = 70% ; VE_SP,2 = 73%. This new parametrisation was used in all analyses presented in the manuscript.

“(3) Regarding these recommendations, when benchmarking the reactive strategies in the sensitivity analysis, comparison should be made between reactive and targeted strategies (i.e., those that specifically target high-transmission environments, regardless of case detection as shown in Fig 2A – green and orange bars). This will ensure a fair comparison of the reactive strategy to alternative static strategies.

In the analysis of alternative scenarios and sensitivity analyses of Figures S4 and S5 we now report the comparison among all strategies.

Logically, for a long enough duration between inoculation and protection, targeted vaccination (or even random vaccination) should be more beneficial than reactive vaccination. This is because the reactive strategy will be too slow and will produce redundant vaccination of individuals who are already infected or recovered. This threshold should be reported so that the limitations of the strategy are clear.

The Reviewer is right that under certain parameters targeted vaccination schemes may be comparable or even more beneficial than reactive vaccination. In the main paper we have considered the high incidence scenario as baseline, consistently with the fact that sustained epidemic activity is currently observed in Europe and the United States (<https://ourworldindata.org/coronavirus>). In this case, the reactive vaccination strategy has a stronger impact with respect to other strategies due to the initial peak in the deployment of

vaccine doses. At a low incidence level, i.e. in the flare-up scenario, few vaccine doses are used at the beginning and the impact of reactive vaccination is more sensitive to the interplay between time scales as mentioned by the Reviewer. In this scenario, we find that assuming a longer delay between the two doses yields a reduced impact of reactive vaccination, up to the point that the advantage with respect to the other strategies is absent.

In Figure 2A, 255 daily vaccinations gives AR RR of about 14% for school targeting, while in Figure S4E, reactive vaccination with a delay of $v_1 = v_2 = 1/14d$ gives AR RR of 17-18%. Therefore, I believe that even a modest relaxation of assumptions as pointed out in (1) and (2) above, towards increased realism, may weaken or reverse the argument favoring reactive vaccination policies.”

The impact of the four strategies depends on the vaccine parameters and the epidemiological context. The Reviewer is right that under certain circumstances the reactive vaccination may provide no advantage over non-reactive strategies. The new version of the manuscript provides a larger exploration of the parameters, allowing understanding under which conditions reactive vaccination is the most effective strategy. In particular, we identified some parameters values for which reactive vaccination produces limited or no advantage with respect to non-reactive strategies. For instance, with the baseline parametrization of the present version of the manuscript - i.e. ~45% of the vaccination coverage at the beginning, a five-week delay to maximum protection on average and vaccine effectiveness as estimated for Delta variant -, we found that for low incidence level, i.e. the flare-up scenario, reactive vaccination produces a relative reduction in the attack rate close to non-reactive strategies.

The main conclusions summarised at the beginning of the Discussion section now read:

-- For a wide range of epidemic scenarios, the reactive vaccination had a stronger impact on the COVID-19 epidemic compared to non-reactive vaccination strategies (including the standard mass vaccination) at equal number of doses used within the two months after inception. In addition, combining reactive and mass vaccination was more effective than mass vaccination alone. For instance, in a scenario of moderate/high incidence with ~45% vaccination coverage at the beginning we found that the relative reduction in the attack rate after two months would improve from 10% to 16% with ~350 daily first vaccine doses per 100000 habitants used in a combined mass/reactive vaccination approach instead of mass only. However, reactive vaccination had limited or no advantage with respect to non-reactive strategies under certain circumstances, as the number of doses administered with the reactive vaccination depended on the number and pace of occurrence and detection of COVID-19 cases. This may be the case when vaccination coverage is already high at the beginning and only a few people to vaccinate are found around detected cases, or in a flare-up scenario when only a few cases are detected. Non-reactive strategies could then be more effective as long as the pace of vaccine administration is not small. Yet, in these situations, adding reactive vaccination to mass vaccination could become of interest again by triggering an increase in vaccine uptake, all the more if this is combined with enhanced TTI.

“Overall, the modelling approach and framework is thorough and well-presented. If the comments above are addressed, I would support publication of the paper in revised form. If addressing the points above weakens or reverses the conclusion with respect to the proposed advantages of reactive vaccination, I would still support publication as long as the result is robust.

Regards,

Cameron Zachreson”

Answer to the comments of Reviewer 2

“The authors here tackle an important problem in COVID-19 epidemiology which is about the efficient use of vaccination to control outbreaks. They use at its basis a detailed multi-layer network model based on known population characteristics based on a a single region (the city of Metz) where the network is dynamic and in simulation, vaccines can be dynamically allocated. Vaccine uptake rates are age dependent but do not appear to otherwise vary.

The model itself has been published and is well established. While the work overall is extensive and informative, I do have concerns about the extent to which the conclusions of the paper are relevant to future of COVID control under current conditions as the assumptions, while undoubtedly merited when this work was originally done, do see to miss some key points about the delta variant in particular. While the basic recommendations themselves about the value of targeted vaccination as a general principle, and the importance of picking up potential clusters of infection (in scenarios where transmission risk is either high and/or important) are well established, even without a modelling analysis, the specifics of the modelling analysis are likely to be more dependent on the characteristics of the virus, as modelled. Thus compromises what modelling can add to the situation in terms of evaluation of the specifics of when, how quickly and with what benefit. Here however, some of the limitations show themselves. as we all know the COVID-19 situation evolves rapidly.”

We thank the Reviewer for pointing out that the work is extensive and informative. The COVID-19 situation indeed is continuously evolving and the emergence of the Delta variant changed substantially the epidemiological landscape. To make the study useful in informing the future course of the COVID-19 epidemic we updated the analysis modifying extensively the parameterization and the scenarios tested. The infection's natural history and the vaccine effect is now parameterized based on available estimates for the Delta variant. We detail the changes in the reply to the dedicated Reviewer's comment below. In addition, other parameters were modified to realistically describe the current situation in terms of virus spread and vaccine deployment. This is detailed in the response to the Editor at the beginning of this point-by-point

reply. A larger set of scenarios is now explored in the main paper and Supplementary Information.

“Here for example, current evidence suggests that the delta variant is substantially more transmissible than previous dominant types (and so is the $R = 1.2$ valid under most future conditions where we are reliant on vaccination to prevent deaths rather than further extensive restrictions), causes different symptoms, and with reduced vaccine efficacy compared to previous variants ([https://www.thelancet.com/journals/lancet/article/PIIS0140-6736\(21\)01290-3/fulltext](https://www.thelancet.com/journals/lancet/article/PIIS0140-6736(21)01290-3/fulltext)); recent evidence also suggests that vaccinated individuals are as likely to reach high viral loads in throat if infected, as unvaccinated individuals (measured through C_t counts - see for example here <https://www.medrxiv.org/content/10.1101/2021.02.16.21251535v3.full.pdf>) and therefore transmit substantially to transmission, but also clear it more efficiently.”

Following Reviewer’s advice we have updated the baseline parameterization based on studies on the Delta variant.

Transmissibility: in all baseline analyses we now consider $R=1.6$, within the range of values estimated at the peak of the Delta wave during summer 2021 (<https://ourworldindata.org/coronavirus> or https://renkulab.shinyapps.io/COVID-19-Epidemic-Forecasting/_w_6f83dd3c/_w_b5ffc5b2/_w_d70a2c2d/_w_9a6450ad/?tab=bag_kt_pred&country=Aargau).

Severity: We reviewed published pre-prints and papers on the severity of the Delta variant. We did not find any new information about the proportion of clinical vs. subclinical symptoms. We thus kept the previous assumption (based on Davies et al Nature Medicine 2020) on the probability of developing clinical symptoms by age. This hypothesis is conservative, since assuming an increase in the probability of clinical symptoms would lead to a higher case detection - clinical cases are defined as showing COVID-19- specific symptoms of moderate to critical intensity - and so a higher impact of reactive vaccination. Several studies showed evidence that infections with the Delta variant have a higher hospitalisation rate - i.e. increased by a multiplicative factor between 1.6 to 2 with respect to Alpha according to the studies ([https://www.thelancet.com/journals/lancet/article/PIIS0140-6736\(21\)01358-1/fulltext](https://www.thelancet.com/journals/lancet/article/PIIS0140-6736(21)01358-1/fulltext), <https://www.thelancet.com/journals/laninf/article/PIIS1473-3099%2821%2900475-8/fulltext>, <https://academic.oup.com/cid/advance-article/doi/10.1093/cid/ciab721/6356459>) . Therefore, in the analysis of hospitalisation, death, etc., outcome reported in the Supplementary Information we decided on an increase in the hospitalisation rate by a multiplicative factor 1.8 (mid range between 1.6 and 2) for all age groups with respect to Alpha. We used the systematic review by Schroeder & al. (<https://www.medrxiv.org/content/10.1101/2021.08.13.21261151v1>) - that estimated an increase in hospitalization rate by a multiplicative factor 1.4 for Alpha with respect to the wild type -, and we obtain an overall relative increase in the risk of hospitalisation by a factor 2.5 ($=1.8*1.4$).

Vaccine Effectiveness: We have now made it clear that we consider a vaccination strategy based on the Cominarty vaccine, that is highly available, can be deployed at a three-weeks

interval between the two doses and is highly effective, all these factors making it the most suitable for reactive vaccination. Estimates of VE for the Delta variant are real life estimates, obtained with different study designs and potentially subject to bias: these are affected by the complex interplay among delta variant emergence, waning of immunity and differential impact by age. We relied on the results of the systematic review by Higdon et al (medRxiv 2021, <https://doi.org/10.1101/2021.09.17.21263549>) which reports VE_{S,1} being between 30% and 65 %, VE_{SP,1} between 35% and 75%, VE_{S,2} between 60% and 80%, and VE_{SP,2} between 55% and 90 %. We set our baseline scenario to the middle of these ranges with VE_{S,1} = 48% ; VE_{SP,1} = 55% ; VE_{S,2} = 70% ; VE_{SP,2} = 73%. This new parameterization was used in all analyses presented in the manuscript. Worse and best case scenarios were also considered, with VE estimates on the lower and upper boundary of the range, respectively.

Time to build immunity following vaccine inoculation: In the original version of the manuscript, the choice of using $v_1=v_2=1/1\text{wk}$ was motivated by the results of the Cominarty phase III trial that showed evidence that approximately 10 days after the first dose the cumulative incidence in the vaccine and Placebo groups diverge (see reply to Reviewer 1). This was also supported by a JCVI statement of January 2021, showing that vaccine efficacy is 90% 2 weeks after the first dose (<https://www.gov.uk/government/publications/prioritising-the-first-covid-19-vaccine-dose-jcvi-statement/optimising-the-covid-19-vaccination-programme-for-maximum-short-term-impact>).

Still, the delay of onset of protection after inoculation may be different for Delta. In many cases VE was assessed 2 weeks after each dose inoculation, i.e. an individual was defined as partially vaccinated starting from 2 weeks after dose 1, and fully vaccinated starting from 2 weeks after dose 2 (<https://www.medrxiv.org/content/10.1101/2021.06.28.21259420v2>). Dosing interval is in general recommended to be 3 weeks - e.g. France, United States, Canada. The Cominarty vaccine was originally authorised at a dosing interval of 3 weeks, and this interval was recommended by, e.g. in Canada, France and United States to contain the epidemic surge due to the Delta variant. Therefore, we made the baseline hypothesis that VE is 0 for the first 2 weeks on average, at an intermediate level for the following period of 3 weeks on average, and at maximum level after.

Infection natural history: we set the incubation period to 5.8 days based on a recently published analysis of a Delta outbreak in China (Kang et al medRxiv <https://doi.org/10.1101/2021.08.12.21261991>). In the same study authors provide evidence that the proportion of presymptomatic transmission is higher for the Delta variant and it is around 74%. We set the value of the average durations of the latent period and the pre-symptomatic transmission to 3.7 days and 2.1 days respectively, to match the 74% estimate. Fraction of pre-symptomatic transmission was computed by analysing the model output with the list of all transmission events, with information of the infection status of the infector.

Viral dynamics: as the Reviewer noted, peak viral load caused by the Delta infection is as high in vaccinated individuals as in unvaccinated ones, suggesting no reduction in infectiousness. This is consistent with the assumptions we made in the original version of the manuscript, i.e. the vaccine confers a reduction in susceptibility, rate of symptoms, but not a reduction in

infectiousness. However, viral clearance was found to be faster, e.g. in (<https://www.medrxiv.org/content/10.1101/2021.02.16.21251535v3.full.pdf>, <https://www.eurosurveillance.org/content/10.2807/1560-7917.ES.2021.26.37.2100824>), suggesting a reduction in infectious period. This reduction was quantified to be around 25% in (<https://www.medrxiv.org/content/10.1101/2021.02.16.21251535v3.full.pdf>). We added this feature in the model. Specifically, we approximately equated clearance duration to the time from onset to recovery and assumed that this period is reduced by 25% in vaccinated individuals.

With this new parametrization we found that results remain qualitative unchanged. With a lower vaccine effectiveness and longer time from inoculation to protection we now find that the benefit of reactive vaccination is slightly reduced compared with the original version of the manuscript. The reactive strategy is more effective than non-reactive strategies, at an equal number of doses, under the majority of parameters and scenarios explored. Yet, to follow the advice of Reviewer 1 and with the aim of better assisting future epidemic control we have now paid attention to describe also the situations where the reactive vaccination may be not interesting compared with non-reactive strategies

The main conclusions summarised at the beginning of the Discussion section now read:

-- For a wide range of epidemic scenarios, the reactive vaccination had a stronger impact on the COVID-19 epidemic compared to non-reactive vaccination strategies (including the standard mass vaccination) at equal number of doses used within the two months after inception. In addition, combining reactive and mass vaccination was more effective than mass vaccination alone. For instance, in a scenario of moderate/high incidence with ~45% vaccination coverage at the beginning we found that the relative reduction in the attack rate after two months would improve from 10% to 16% with ~350 daily first vaccine doses per 100000 habitants used in a combined mass/reactive vaccination approach instead of mass only. However, reactive vaccination had limited or no advantage with respect to non-reactive strategies under certain circumstances, as the number of doses administered with the reactive vaccination depended on the number and pace of occurrence and detection of COVID-19 cases. This may be the case when vaccination coverage is already high at the beginning and only a few people to vaccinate are found around detected cases, or in a flare-up scenario when only a few cases are detected. Non-reactive strategies could then be more effective as long as the pace of vaccine administration is not small. Yet, in these situations, adding reactive vaccination to mass vaccination could become of interest again by triggering an increase in vaccine uptake, all the more if this is combined with enhanced TTI.

"The assumption that all symptomatic individuals get picked up is quite optimistic: this is evident in the UK data which I am more familiar with due to the high number of individuals who are tested and are positive when entering hospital - i.e. not nosocomial infections but likely individuals who went to hospital because they were seriously ill and only tested positive then. Thus there are likely to be many people who have relatively mild infections, who never get

tested. Comparisons to random surveillance data suggests that the number may be as low as one in four ()”

We have assumed that, as a result of the routine TTI in place in France, 50% of clinical cases and 10% of subclinical cases would be detected. Note that we used the definition of clinical/subclinical cases of Davies et al (<https://doi.org/10.1038/s41591-020-0962-9>) and Riccardo et al (<https://doi.org/10.2807/1560-7917.ES.2020.25.49.2000790>), according to which clinical cases have COVID-19-specific symptoms with moderate to critical intensity, while subclinical cases have none to mild and nonspecific symptoms. We computed the overall fraction of infectious (clinical and subclinical combined) that is detected with this parameterization and we found that ~25% of cases are detected, close to the estimate pointed by the Reviewer, and in line with estimates for France in 2020 (Pullano et al. Nature 2021, Hozé et al Lancet Public Health) - we have now explicitly mentioned this aspect in the text. In the analysis of the emergence of a variant of concern (Figure 4 in the main paper) we also considered a scenario of enhanced TTI. This represents a situation of alert, where case investigation, screening and sensibilisation campaigns are implemented in the affected territory, resulting in a higher case detection, more contact identified and increased adherence to isolation. In the original version of the manuscript we set 100% and 50% detection for clinical and subclinical cases, respectively, in this enhanced TTI scenario. This yields to a ~70% detection rate overall. Prompted by the Reviewer comment we have now lowered this value to 70% and 30% for probability of the detection of clinical and subclinical cases, respectively. This corresponds to a ~45% detection rate overall. Unfortunately, we are not aware of any information source that can be used to inform this parameter choice in a variety of countries in the current situation. However, we report in the paper the comparison between enhanced and baseline TTI, i.e. two extreme situations, which helps understanding the impact of this ingredient on reactive vaccination effectiveness.

“Reduction of transmission in the models, is based on google mobility data - it is unclear how well correlated these two factors are - certainly in the UK, mobility has been changing at rates that seem to have little correlation to either infection rates or estimates of R, and are quite different from contact rates estimated by the "COMIX" survey (<https://bmcmmedicine.biomedcentral.com/articles/10.1186/s12916-020-01597-8>) and its the latter which seems to correlate well with changes in R.”

We agree with the Reviewer that the relationship between mobility as estimated by Google mobility reports and COVID-19 transmission potential is not simple to capture. The epidemic transmission potential is determined by multiple factors, including social contacts and the adoption of barrier measures. We used Google mobility reports to coarsely inform the repartition of contacts across the different settings. This is supported by a previous study showing that informing contact matrices on the basis of mobility variations in different settings better described the epidemic trajectory (Pullano et al. Nature 2021). However, the reproductive ratio was parameterized independently from the choice of such a repartition. Specifically, once defined the level of teleworking and social distancing, we have set the transmission rate per contact to recover the desired reproductive ratio, by following the

procedure explained in the section “Details on the epidemic simulations” of the Supplementary Information. We realised that this was not clear in the original version of the manuscript we have thus added a sentence in the Method section to explain this point:

--Importantly, the reproductive ratio is set to the desired value, independently by the level of contact reduction, as described in the Supplementary Information.

We stress that a precise calibration of the repartition of contacts across settings due to teleworking/social distancing was not our objective. All the more that human activity has varied greatly since the vaccination campaign started and will likely vary in the near future in response to the epidemic situation and the possible implementation of restrictions.

In the baseline scenarios we now account for limited social restrictions compared with the original version of the paper. Specifically, we have set a 5% reduction in the contacts occurring in the community and a proportion of teleworking equal to 10%. These values are the one reported for France by google mobility reports for October 2021. They are also in the middle of the range of estimates for Europe. Interestingly, a proportion of individuals teleworking close to 10% is also reported by Yougov (<https://yougov.co.uk/topics/international/articles-reports/2020/03/17/personal-measures-taken-avoid-covid-19>) for France and other European countries. In the sensitivity analysis reported in the Supplementary Information we compared different teleworking/social distancing scenarios at equal value of R, finding that this has limited impact on the results.

“I would stress that the analysis itself seems solid and is based on an established model. However, because of the ever changing COVID situation, I do not see it as being of important general interest - the general aspects are good but not novel while the individual analyses have novelty but aren't currently relevant.”

The new version of the manuscript accounts for delta-specific parameters and considers epidemic scenarios that better reflect the current epidemic situation in Western countries. In addition, the manuscript now includes the exploration of a larger range of parameters and different scenarios. We argue that, with regard to the ever changing COVID19 situation, the large exploration of parameters of uncertain value allow precisising the conditions that make reactive vaccination of interest and can serve as a base for informed decision making on the best allocation of resources.

Reviewer #3 (Remarks to the Author):

“The authors built an individual-based model to explore the impact of different vaccination strategies, especially the reactive vaccination of workplaces and schools. I believe this is a novel study that is almost ready to publish. Before that, some issues need to be addresses:”

We thanks the reviewer for pointing out the novelty of the study

“1. In model shown in Figure 3C, table S3 and table S4, the probability of detecting a subclinical case is set as .1 and .5 for both unvaccinated infectious and vaccinated infectious. I assume for vaccinated subclinical cases, the probability of detecting is much smaller than the clinical cases. I wonder, whether the heterogeneity of probability in unvaccinated and vaccinated subclinical cases makes a huge difference? It seems that the Delta Variant breakthrough infections spreading is common (even if with a shorter infectious period). If the vaccinated subclinical transmission is common, I imagine the surveillance will be very effective to curb the Delta spreading.”

Indeed when a large proportion of the population is already vaccinated breakthrough infection covers a major role with a larger proportion of subclinical cases and in turn a reduced detection rate overall. This makes the detection of outbreaks more difficult, thus hinders the implementation of the reactive vaccination. We added the following sentence in the Discussion section:

-- [...] breakthrough infection becomes an important driver of propagation with consequently a larger proportion of subclinical cases and in turn a reduced detection rate overall. This makes the detection of outbreaks more difficult.

“2. I had a hard time understanding Figure 2.

(a) In panel B legend, is "Random" the same as "Mass" in other panels?

(b) In line 133-134, "Figure 2B" should be consistent with "Panel A".

(c) The description for Figure 2 panel A-B is a bit back-and-forth: the authors first briefly described panel A and panel B then only talked about panel A. Can the authors talk more about panel B?”

We thank the reviewer for noticing this. In modifying Figure 2, as described in the reply to the Editor comment, we have paid attention to fix these issues and improve the clarity of the figure and its description.

REVIEWERS' COMMENTS

Reviewer #1 (Remarks to the Author):

The paper has been thoroughly revised, taking into account additional scenarios, and making a much more balanced appraisal of reactive vaccination as a COVID-19 mitigation strategy. The response to my comments on the first draft was extremely thorough and I appreciate the level of detail provided in the response letter.

I support publication of the manuscript after some relatively minor revisions:

1) The paper should be revised to improve clarity. This applies to most of the descriptions of scenarios and analysis processes, in which subsets of the complex parameter space are chosen for comparison. The parameter choices and the reasons for them (in terms of the comparisons being made) need to be more clear. In each of the figures and figure captions, care should be taken that all axis labels and legends are provided and are clearly interpretable to the reader. For example, red trajectories in Fig. 3(c,e) are not shown in the legends. As another example, there is no legend provided to differentiate the green and black points in Figure 4 (I assume this corresponds to the legend in figure S9). Also, many of the axis labels use the syntax "...x100,000", I think this is meant to mean 'per 100,000 inhabitants' but reads like a scaling factor. This should be corrected throughout. Overall, I found myself doing a lot of work to 'connect the dots', I believe all of the necessary analysis is shown to support the conclusions and discussion, but clarity can be improved to reduce the burden to the reader.

2) the paper should be examined for grammatical errors, it mostly reads well but there are some confusing word choices (i.e., in the abstract, the sentence "few people are found to vaccinate around cases" is confusing - it reads as though it describes a 'finding' but is referring to the action of the intervention model in 'locating' individuals to vaccinate.

Noting here that the R value of 1.6 could be interpreted as a low estimate depending on what it is meant to correspond to (i.e., is this the 'fundamental' transmission rate of the virus, or the mitigated transmission rate due to ongoing social measures and behavioural changes?). In my opinion this specific choice seems realistic in the chosen context, but perhaps some justification for this choice and some further explanation would help readers understand the parameter, as the number is substantially lower than most 'raw' estimates of R0 for any of the SARS-CoV-2 lineages that have been analysed so far.

Reviewer #2 (Remarks to the Author):

The authors have done a considerable amount of work to update their manuscript and these have definitely made it a more generally relevant paper.

I have a few remaining questions.

i) Mass vaccination always performs worse than other strategies in their evaluations. this is, I assume because of the nature of the assumptions that are made about it (in an extreme scenario, if you were able to vaccinate everyone at the same per person rate everywhere as a targeted strategy, with no constraints on supply, this should work at least as well in reducing incidence, but of course may be relatively inefficient. it should be better if the targeted strategies are sufficiently 'leaky' (i.e. the targeting is good). Some exploration of this would be useful (even as a discussion point).

ii) The incidence curves are shown to about 7 weeks, uptake and others to 60 days - it would be helpful to be consistent and at least in some instances, get a sense of what the overall trajectory is like - are there some advantages in terms of rapidity of decline for example that are not shown here?

iii) What is the distribution of outcomes across simulations. If the distributions are very similar across strategies, then the mean values are probably good enough for decision-making. However if they are different (e.g. greater variability for reactive strategies, compared to non-reactive) there may be trade-offs associated with the likelihood of a more severe outcome, as opposed to just the mean values, that may need to be considered.

Answer to comments of Reviewer 1

The paper has been thoroughly revised, taking into account additional scenarios, and making a much more balanced appraisal of reactive vaccination as a COVID-19 mitigation strategy. The response to my comments on the first draft was extremely thorough and I appreciate the level of detail provided in the response letter.

I support publication of the manuscript after some relatively minor revisions:

We thank the Reviewer for recognising that the manuscript is improved and for his additional comments. We provide here below a point-by-point reply to Reviewer remarks.

1) The paper should be revised to improve clarity. This applies to most of the descriptions of scenarios and analysis processes, in which subsets of the complex parameter space are chosen for comparison. The parameter choices and the reasons for them (in terms of the comparisons being made) need to be more clear. In each of the figures and figure captions, care should be taken that all axis labels and legends are provided and are clearly interpretable to the reader. For example, red trajectories in Fig. 3(c,e) are not shown in the legends. As another example, there is no legend provided to differentiate the green and black points in Figure 4 (I assume this corresponds to the legend in figure S9). Also, many of the axis labels use the syntax "...x100,000", I think this is meant to mean 'per 100,000 inhabitants' but reads like a scaling factor. This should be corrected throughout. Overall, I found myself doing a lot of work to 'connect the dots', I believe all of the necessary analysis is shown to support the conclusions and discussion, but clarity can be improved to reduce the burden to the reader.

We revised the paper for clarity of presentation. In particular, we now define the baseline analysis in detail in the Results section and highlight the differences with the other scenarios explored as they are introduced. The captions of the Figures contain now more details on the parameters and better distinguish between the baseline and the other scenarios explored.

We apologize for the missing legends and thank the Reviewer for spotting them. Missing legends and unclear axes' labels in the figures were revised according to the Reviewer's suggestions. Additional revisions in the figures were made as requested by the editor.

2) the paper should be examined for grammatical errors, it mostly reads well but there are some confusing word choices (i.e., in the abstract, the sentence "few people are found to vaccinate around cases" is confusing - it reads as though it describes a 'finding' but is referring to the action of the intervention model in 'locating' individuals to vaccinate.

We have thoroughly revised the paper and hope that the writing is now clearer.

Noting here that the R value of 1.6 could be interpreted as a low estimate depending on what it is meant to correspond to (i.e., is this the 'fundamental' transmission rate of the virus, or the mitigated transmission rate due to ongoing social measures and behavioural changes?). In my opinion this specific choice seems realistic in the chosen context, but perhaps some justification for this choice and some further explanation would help readers understand the parameter, as the number is substantially lower than most 'raw' estimates of R0 for any of the SARS-CoV-2 lineages that have been analysed so far.

The reproductive ratio used in our analysis corresponds with an effective reproduction ratio integrating the effect of interventions and the level of disease and vaccine induced immunity in the population. As such it should not be compared with 'raw' estimates of R_0 , but instead with effective reproductive ratio estimates (R_t index) measured during the outbreak. In particular, the value of 1.6 was chosen, being this in the range of values estimated at the peak of the Delta wave during summer 2021, as discussed in the Results section of the main paper.

We understand that this is not clearly explained in the text, thus we added in the Supplementary Information the following sentence.

– Therefore, it (the reproductive number R) integrates the effect of the interventions and the level of disease and vaccine induced immunity in the population at the start.

Answer to comments of Reviewer 2

The authors have done a considerable amount of work to update their manuscript and these have definitely made it a more generally relevant paper.

We thank the reviewer for this nice assessment.

I have a few remaining questions.

We have carefully addressed the additional comments as detailed below.

i) Mass vaccination always performs worse than other strategies in their evaluations. this is, I assume because of the nature of the assumptions that are made about it (in an extreme scenario, if you were able to vaccinate everyone at the same per person rate everywhere as a targeted strategy, with no constraints on supply, this should work at least as well in reducing incidence, but of course may be relatively inefficient. it should be better if the targeted strategies are sufficiently 'leaky' (i.e. the targeting is good). Some exploration of this would be useful (even as a discussion point).

We agree with the Reviewer that this point could be explored further. However, we suggest that a systematic assessment of the assumptions and their implications is beyond the scope of this work. The paper is already dense and provides an extensive study of several key factors. We believe that including even more would make the paper very hard to follow. Notwithstanding, we acknowledged this point in the Results section.

– Among the three strategies, the reduction produced by mass vaccination was slightly lower. This is because the strategies are compared at the same number of daily vaccine doses and, in workplaces/universities and school locations, these doses were directed to a more active population - working population, or population living in large households - with a greater potential to transmit the infection.

ii) The incidence curves are shown to about 7 weeks, uptake and others to 60 days - it would be helpful to be consistent and at least in some instances, get a sense of what the overall trajectory

is like - are there some advantages in terms of rapidity of decline for example that are not shown here?

While the same time frame is used in all analyses (2 months or 60 days), it is true that we chose to plot incidence of clinical cases by week but vaccine doses by day, leading to this perceived lack of consistency. This was done to ease comparison with real world data, where incidence is often reported weekly while the number of vaccines used is reported daily, since it is directly linked to logistical efforts. We would prefer to leave it this way, even if we acknowledge that this comes at a price.

Please note that we have chosen throughout the paper not to expand the analysis further than two months as we expect that the kind of reactive strategy studied here would not be implemented for long in real life in western countries. Therefore, the effect of vaccination on the decline after the peak is therefore not studied here.

iii) What is the distribution of outcomes across simulations. If the distributions are very similar across strategies, then the mean values are probably good enough for decision-making. However if they are different (e.g. greater variability for reactive strategies, compared to non-reactive) there may be trade-offs associated with the likelihood of a more severe outcome, as opposed to just the mean values, that may need to be considered.

We explored the distribution of the attack-rate, comparing all vaccination strategies, considering as an example the baseline scenario analysed in Fig. 2e. The analysis is now provided in Supplementary Figure 3. We found similar levels of dispersion, hence the results differed in location rather than in scale. This supports the choice to compare mean values throughout the paper.